# Stress-activated MAPK signaling controls fission yeast actomyosin ring integrity by modulating formin For3 levels

**Elisa Gómez-Gil[1], Rebeca Martín-García[2], Jero Vicente-Soler[1], Alejandro Franco[1], Beatriz Vázquez-Marín[1], Francisco Prieto-Ruiz[1], Teresa Soto[1], Pilar Pérez[2], Marisa Madrid[1]\*, Jose Cansado[1]\***

[1]Yeast Physiology Group, Departamento de Genética y Microbiología, Facultad de Biología. Universidad de Murcia, Murcia, Spain; [2]Instituto de Biología Funcional y Genómica (IBFG), Consejo Superior de Investigaciones Científicas, Universidad de Salamanca, Salamanca, Spain

**Abstract** Cytokinesis, which enables the physical separation of daughter cells once mitosis has been completed, is executed in fungal and animal cells by a contractile actin- and myosin-based ring (CAR). In the fission yeast *Schizosaccharomyces pombe,* the formin For3 nucleates actin cables and also co-operates for CAR assembly during cytokinesis. Mitogen-activated protein kinases (MAPKs) regulate essential adaptive responses in eukaryotic organisms to environmental changes. We show that the stress-activated protein kinase pathway (SAPK) and its effector, MAPK Sty1, downregulates CAR assembly in *S. pombe* when its integrity becomes compromised during cytoskeletal damage and stress by reducing For3 levels. Accurate control of For3 levels by the SAPK pathway may thus represent a novel regulatory mechanism of cytokinesis outcome in response to environmental cues. Conversely, SAPK signaling favors CAR assembly and integrity in its close relative *Schizosaccharomyces japonicus,* revealing a remarkable evolutionary divergence of this response within the fission yeast clade.

**\*For correspondence:**
marisa@um.es (MM);
jcansado@um.es (JC)

**Competing interests:** The authors declare that no competing interests exist.

## Introduction

Accurate cell division in eukaryotes requires the physical separation of daughter cells once mitosis has been completed. This process, known as cytokinesis, is a conserved mechanism that relies on the formation of a ring made of actin filaments and class II myosins (contractile actomyosin ring or 'CAR'), which constricts and allows daughter cell separation by plasma membrane fusion (*Gerien and Wu, 2018*; *Pollard and Wu, 2010*; *Mangione and Gould, 2019*). Cytokinesis is a strictly fine-tuned biological process whose failure may lead to aneuploidy or multi-nucleate cells, resulting in cell death or tumorigenesis (*Sagona and Stenmark, 2010*). The rod-shaped fission yeast *Schizosaccharomyces pombe* is a popular model organism for the study of cytokinesis due to its small size, a relatively simple actin-based cytoskeletal network (actin patches, actin cables, and the CAR), and an easily tractable small genome with little gene redundancy (*Rincon and Paoletti, 2016*; *Balasubramanian et al., 2004*). These features have allowed the identification of a large number of evolutionary conserved cytokinesis proteins and the precise spatiotemporal analysis of cytokinetic events (*Pollard and Wu, 2010*).

In *S. pombe*, the development of a mature CAR takes place early after cytokinetic factors assemble into medial precursor nodes during the mitotic onset (*Pollard and Wu, 2010*). *S. pombe* starts to assemble the CAR in metaphase when chromosomes are not yet fully segregated, and unlike most animal and fungal cells including those of *S. japonicus,* another fission yeast species (*Gu and Oliferenko, 2015*). Once the nuclear division is completed, the CAR constricts and drives plasma

membrane closure, and the concomitant deposition of a cell wall septum whose degradation results in the physical separation of daughter cells (*Pollard and Wu, 2010*; *Rincon and Paoletti, 2016*; *Balasubramanian et al., 2004*). The activity of the septation initiation network (SIN), a GTPase-protein kinase signaling cascade, secures the precise coordination between mitosis and cytokinesis, and prevents the later to occur before chromosomes have safely segregated (*Roberts-Galbraith and Gould, 2008*; *Simanis, 2015*).

The formin Cdc12 is essential for the nucleation and elongation of actin filaments from free actin monomers during CAR assembly in this organism (*Chang et al., 1997*; *Kovar et al., 2003*), and its activity is tightly regulated post-translationally through phosphorylation (*Bohnert et al., 2013*; *Willet et al., 2018*). Later, and upon full activation, the SIN terminal efector kinase Sid2 phosphorylates Cdc12 to inhibit its multimerization and facilitate mature CAR assembly and maintenance (*Bohnert et al., 2013*). Conversely, Cdc12 phosphorylation by cyclin-dependent kinase-1 (Cdk1/Cdc2) during mitosis antagonizes Cdc12 localization at the CAR and delays its formation (*Willet et al., 2018*).

For3 is a non-essential diaphanous-like *S. pombe* formin that assembles actin cables for cellular transport and has established roles in polarized secretion and growth during interphase (*Feierbach and Chang, 2001*; *Martin and Chang, 2006*; *Martin et al., 2007*; *Scott et al., 2011*). In the absence of For3 cells maintain a polarized growth, but show altered morphology and present a defect in NETO ('New End Take Off') (*Martin et al., 2005*). For3 neither localizes to the cell poles nor binds to its cortical tethers when found in an autoinhibited (closed) state, which is mediated by an intramolecular interaction between the autoregulatory (DAD) and inhibitory (DID) domains (*Martin et al., 2007*). The switch of For3 from an autoinhibited to an active (open) conformation and its localization is triggered by the active form of the essential Rho-GTPase Cdc42 (*Martin et al., 2007*). Although not essential for cytokinesis (*Martin and Chang, 2006*), For3 also localizes to the division site, arriving shortly after SPB separation during CAR assembly and before the spindle begins to elongate during anaphase B (*Coffman et al., 2013*). The later stages of CAR constriction and disassembly are delayed in *for3Δ* cells as compared to wild-type cells (*Coffman et al., 2013*). Accordingly, For3-nucleated actin cables are critical for efficient delivery of secretory vesicles by myosin-V Myo52 to the division site (*Lo Presti et al., 2012*; *Wang et al., 2016*). While For3 function is only required during CAR constriction in wild-type cells, it becomes essential during ring assembly for node movement and cell survival in mutants with lower Cdc12 activity (*Coffman et al., 2013*). This suggests that For3 cooperates with Cdc12 to nucleate actin filaments for CAR assembly and disassembly during cytokinesis (*Coffman et al., 2013*). However, the regulatory mechanism/s involved in such function has yet to be fully disclosed.

The conserved mitogen-activated protein kinase (MAPK) signaling pathways including the extracellular signal-regulated kinase (ERK) and p38 class, play a key role in higher eukaryotes to elicit adaptive responses to changes in the external and internal cellular environment that include the dynamic remodeling of the actin cytoskeleton, which is crucial for cell motility and migration (*Pullikuth and Catling, 2007*; *Pullikuth and Catling, 2010*; *Mendoza et al., 2011*; *Cuenda, 1773*; *Tanimura and Takeda, 2017*). However, the cellular targets and multiple mechanisms employed by MAPKs to modulate actin organization are far from being clarified. The stress-activated protein kinase pathway (SAPK) and its central element, the MAPK Sty1 (a p38 ortholog), plays a critical role in *S. pombe* by controlling cell cycle progression and the general response to adverse situations including hyperosmotic stress, heat shock, hydrogen peroxide, or nutrient starvation (*Pérez and Cansado, 2010*; *Gacto et al., 2003*). The SAPK pathway is also activated when cells are treated with the sponge-derived macrolide Latrunculin A (LatA) (*Mutavchiev et al., 2016*), which sequesters actin monomers to prevent F-actin cytoskeleton assembly and also accelerates actin filament depolymerization (*Morton et al., 2000*; *Fujiwara et al., 2018*). Activated Sty1 causes the dispersal of active Cdc42 from the cell poles and the cessation of polarized growth (*Mutavchiev et al., 2016*).

In this work, we show that the SAPK pathway negatively controls CAR assembly in *S. pombe* by promoting a specific reduction in For3 levels in response to actin cytoskeleton damage induced with LatA. Remarkably, SAPK signaling instead contributes positively to CAR assembly during actin stress in *S. japonicus.* The opposite role played by the SAPK pathway during control of CAR integrity likely reflects the evolutionary adaptation of this signaling cascade to the marked differences in the execution of cytokinesis in both fission yeast species.

## Results

### The SAPK pathway becomes activated and negatively regulates fission yeast growth in the presence of latrunculin A

LatA prevents F-actin cytoskeleton assembly in eukaryotic organisms (*Morton et al., 2000*). Treatment of growing cells of the fission yeast *S. pombe* with high concentrations of LatA (>10 µM) elicits disassembly of patches, cables, and the CAR (*Pelham and Chang, 2001*; *Karagiannis et al., 2005*), and is accompanied by strong activation of Sty1 (*Mutavchiev et al., 2016*), the core MAPK of the SAPK pathway (*Figure 1A and C*). *S. pombe* wild-type cells were unable to grow in YES plates supplemented with low concentrations of LatA (0.15–0.2 µM), which disassemble the cables and the CAR, but not the patches (*Figure 1B*; *Karagiannis et al., 2005*; *Tournier et al., 2004*; *Asadi et al., 2016*). Unexpectedly, mutant cells lacking Sty1 grew in these low LatA concentrations (*Figure 1B*). Null mutants in upstream elements of this signaling cascade shared this phenotype, which includes the response regulator Mcs4, the redundant MAPKKK´s Wak1 and Win1, and MAPKK Wis1 (*Figure 1A*, *Figure 1—figure supplement 1*; *Pérez and Cansado, 2010*). Conversely, cells expressing the constitutively active MAPKK allele *wis1DD* that increases basal Sty1 activity (*Shiozaki and Russell, 1997*), or mutants lacking the respective Sty1 tyrosine and serine/threonine phosphatases Pyp1 and Ptc1 (*Figure 1A*; *Millar et al., 1995*; *Nguyen and Shiozaki, 1999*), did not grow in LatA, and this phenotype was totally suppressed in both *pyp1Δ sty1Δ* and *ptc1Δ sty1Δ* double mutants (*Figure 1B*).

Treatment of wild-type *S. pombe* cells for 1 hr with a range of concentrations of LatA, including those that inhibit growth in rich medium (0.2 µM), activated Sty1 as measured by western blot analysis with an anti-phospho-p38 antibody (*Figure 1C*). The magnitude and dynamics of MAPK activation during LatA treatment were enhanced (~2 to 5X times) and accelerated, respectively, in a dose-dependent manner (*Figure 1D*). The maximal ratio of Sty1 phosphorylation in *S. pombe* is achieved in response to a saline osmotic stress, and results in a ~ 20X increase in activation units with respect to the basal levels (*Prieto-Ruiz et al., 2020*). Thus, the lower activation threshold reached after LatA treatment may represent a relatively small fraction of Sty1 being phosphorylated (~10% to 25%). Sty1 activation induced with LatA was absent in cells expressing a LatA-resistant actin mutant allele (*Act1-LQ*) (*Figure 1—figure supplement 2*; *Karagiannis et al., 2005*), confirming that MAPK activation is due to its specific deleterious effects on actin and/or actin-based networks. Cells expressing a mutant allele Mcs4(*D512N*) that does not activate the SAPK pathway upon stimulation with hydrogen peroxide (*Shieh et al., 1997*), displayed Sty1 activation during LatA treatment, while this response was absent in a *mcs4Δ* mutant (*Figure 1E*). Hence, like other environmental cues, the signal induced by LatA treatment is transduced to the SAPK MAPK module via Mcs4 response regulator, and is independent on the function of the two-component system that functions explicitly in response to hydrogen peroxide-mediated oxidative stress (*Pérez and Cansado, 2010*). *sty1Δ* cells expressing the dually phosphorylated (wild type) MAPK at tyrosine and threonine, but not the mono-phosphorylated versions, were growth-sensitive to LatA (*Figure 1—figure supplement 3*). Therefore, as for other biological roles (*Vázquez et al., 2015*), modulation of LatA growth by Sty1 requires a fully activated MAPK. Growth in plates with LatA of cells expressing a nuclear-excluded, constitutive plasma membrane-targeted version of Sty1 (Sty1-GFP-CAAX) was quite similar than in control cells (Sty1-GFP) (*Figure 1—figure supplement 4*), suggesting that Sty1-negative control of cell growth in the presence of LatA is mainly exerted in a transcription-independent fashion. Indeed, the transcription factor Atf1, which becomes phosphorylated by Sty1 during stress to induce the expression of the Core Environmental Stress Response genes (CESR) (*Pérez and Cansado, 2010*; *Shiozaki and Russell, 1996*; *Wilkinson et al., 1996*; *Chen et al., 2003*), became activated also in response to LatA, as shown by its mobility shift during western blot analysis (*Figure 1C*), but, opposite to *sty1Δ* cells, *atf1Δ* cells showed lower tolerance to LatA (*Figure 1F*).

Although high concentrations of LatA (>10 µM) induce dispersal from the cell poles of the activated Rho GTPase Cdc42 in a SAPK-dependent fashion (*Mutavchiev et al., 2016*), low drug concentrations (0.2 µM) did not elicit such response, as determined by monitoring the subcellular localization of a GFP–tagged Cdc42/Rac interactive-binding peptide probe (CRIB-3xGFP) that specifically detects the activated state of the GTPase (*Figure 1G*; *Tatebe et al., 2008*). The time needed for dispersal of the CRIB-3xGFP fusion from the cell poles after initial SPB separation was nearly identical in wild-type cells during unperturbed growth and in 0.15 µM LatA-treated

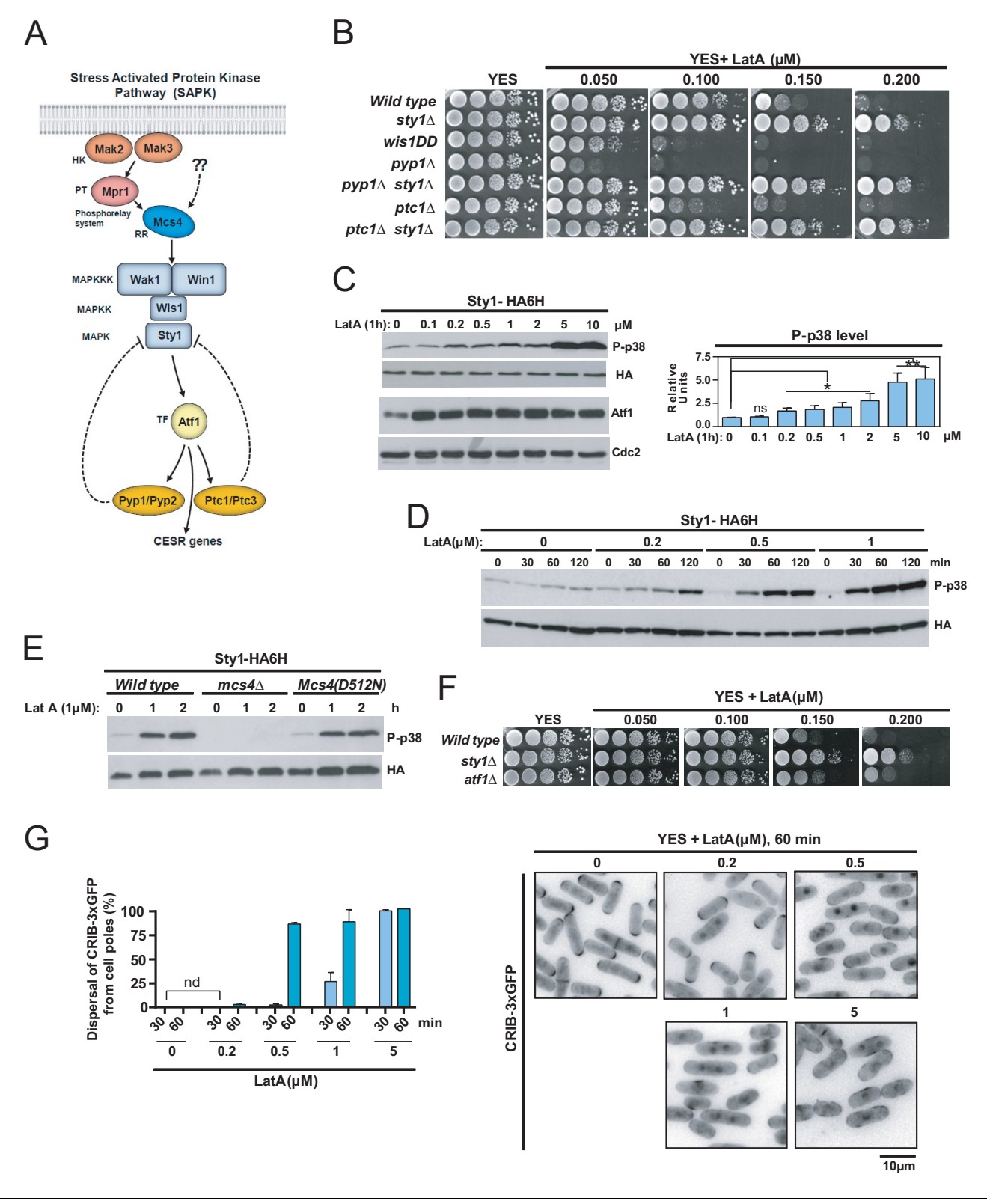

**Figure 1.** The SAPK pathway becomes activated and negatively controls *S. pombe* growth in the presence of low Latrunculin A concentrations. (**A**) The stress activated MAPK pathway (SAPK) in *S. pombe*. Please see text for a detailed description of its main components and functions. (**B**) Decimal dilutions of strains of the indicated genotypes were spotted on YES and YES solid plates with a range of concentrations of LatA, incubated at 30°C for 3 days, and photographed. A representative experiment is shown. (**C**) Left panel: *S. pombe* wild type cells expressing a genomic Sty1-HA6his fusion were

*Figure 1 continued on next page*

*Figure 1 continued*

grown in YES medium to mid-log phase, and remained untreated (0), or treated with a range of concentrations of LatA for 1 hr. Activated/total Sty1 were detected with anti-phospho-p38 and anti-HA antibodies, respectively. Total Atf1 levels were detected with anti-Atf1 antibody. Anti-Cdc2 was used as a loading control. Right panel: Relative units as mean ± SD (biological triplicates) for Sty1 phosphorylation (anti-phospho-p38 blot) were determined with respect to the internal control (anti-HA blot). **, p<0.005; *, p<0.05; ns, not significant, as calculated by unpaired Student's *t* test. (D) *S. pombe* wild-type cells expressing a genomic Sty1-HA6his fusion were grown in YES medium to mid-log phase, and remained untreated (0), or treated with a range of concentrations of LatA for the indicated times. Activated/total Sty1 were detected with anti-phospho-p38 and anti-HA antibodies, respectively. (E) *S. pombe* wild type, *mcs4Δ* and Mcs4(*D512N*) cells expressing a genomic Sty1-HA6his fusion were grown in YES medium to mid-log phase, and treated with 1 µM LatA for the indicated times. Activated/total Sty1 were detected with anti-phospho-p38 and anti-HA antibodies, respectively. A representative experiment is shown. (F) Decimal dilutions of strains of the indicated genotypes were spotted on YES and YES solid plates with a range of concentrations of LatA, incubated at 30°C for 3 days, and photographed. (G) *S. pombe* wild-type cells expressing a CRIB-3xGFP fusion were grown in YES medium to mid-log phase, and remained untreated (0) or treated with a range of concentrations of LatA for 30 and 60 min. Left panel: The percentage of cells at G1 and G2 that show dispersal of the CRIB-3xGFP fusion from the cell poles was estimated in each case by fluorescence microscopy, and is presented as mean ± SD (biological duplicates). nd: no dispersal from the cell poles is detected. Right panel: representative fluorescence micrographs of control and LatA-treated cells.

The online version of this article includes the following source data and figure supplement(s) for figure 1:

**Source data 1.** Values used for graphical representations and statistical analysis in *Figure 1*.
**Figure supplement 1.** Growth of SAPK null mutants in the presence of LatA.
**Figure supplement 2.** Sty1 activation with LatA in wild type cells *vs* a LatA-insensitive mutant.
**Figure supplement 3.** Growth of Sty1 phosphorylation mutants in the presence of LatA.
**Figure supplement 4.** Effect of LatA on growth of a plasma membrane-bound Sty1 mutant.
**Figure supplement 5.** Dispersal of active Cdc42 from the cell poles in the presence or absence of LatA.
**Figure supplement 6.** Sty1 deletion alleviates LatA-sensitivity of mutants lacking Cdc42 GEF's.

cultures (~11–12 min; *Figure 1—figure supplement 5*). Additionally, Sty1 deletion partially suppressed defective growth with LatA of mutant strains lacking Cdc42 GEFs Gef1 or Scd1 (*Figure 1—figure supplement 6*). Altogether, these evidences suggest that inhibition of fission yeast growth by Sty1 in the presence of low LatA concentrations is not funneled through changes in Cdc42 activity.

## SAPK activity negatively regulates CAR integrity in response to latrunculin A

The growth phenotypes of lack- and gain-of-function SAPK mutants in the presence of LatA might result from alterations in the organization and/or dynamics of the actin cytoskeleton. mRNA expression levels of the actin encoding gene (*act1+*) (*Figure 2—figure supplement 1*), and Act1 protein levels (*Figure 2—figure supplement 2*), were virtually identical in wild type, *sty1Δ*, and *wis1DD* mutant cells, indicating that the SAPK pathway does not affect actin gene mRNA expression and protein synthesis.

By employing confocal fluorescence microscopy of strains expressing the actin probe Pact1-LifeAct-GFP fusion (*Huang et al., 2012*), we found that the integrity of actin patches and cables was not significantly altered in *sty1Δ* and *wis1DD* mutants as compared to wild type cells during unperturbed exponential growth (*Figure 2A*). The number/density of actin patches per cell did not vary in exponentially growing cultures of the above strains treated with 0.15 µM LatA (*Figure 2A*). These results confirmed that LatA affects the rate of actin polymerization in a concentration-dependent manner (*Rupes et al., 1999*), and that higher concentrations of this drug (>10 µM) are required to achieve complete depolymerization of F-actin patches (*Karagiannis et al., 2005*; *Tournier et al., 2004*). However, patches partially depolarized in 0.15 µM LatA-treated wild type and *wis1DD* cells, as evidenced by an increase in the number of patches located in the central third of the longitudinal cell axis (*Figure 2A and B*). Contrariwise, they remained mostly polarized during the same LatA treatment in the *sty1Δ* mutant (*Figure 2A and B*). The time taken for complete depolymerization of F-actin patches with high concentrations of LatA (20 µM) was very similar in wild type and *sty1Δ* cells (*Figure 2—figure supplement 3*), suggesting that LatA-resistance in this mutant is not related to differences in drug entry induced by MAPK deletion. The number of cells with cables during unperturbed growth was maximal (~100%) and did not change significantly between wild-type, *sty1Δ*, and *wis1DD* cells (*Figure 2C*). Consistent with previous reports (*Tournier et al., 2004*), the number of cells with visible actin cables was reduced in wild-type cells after treatment with 0.15 µM LatA

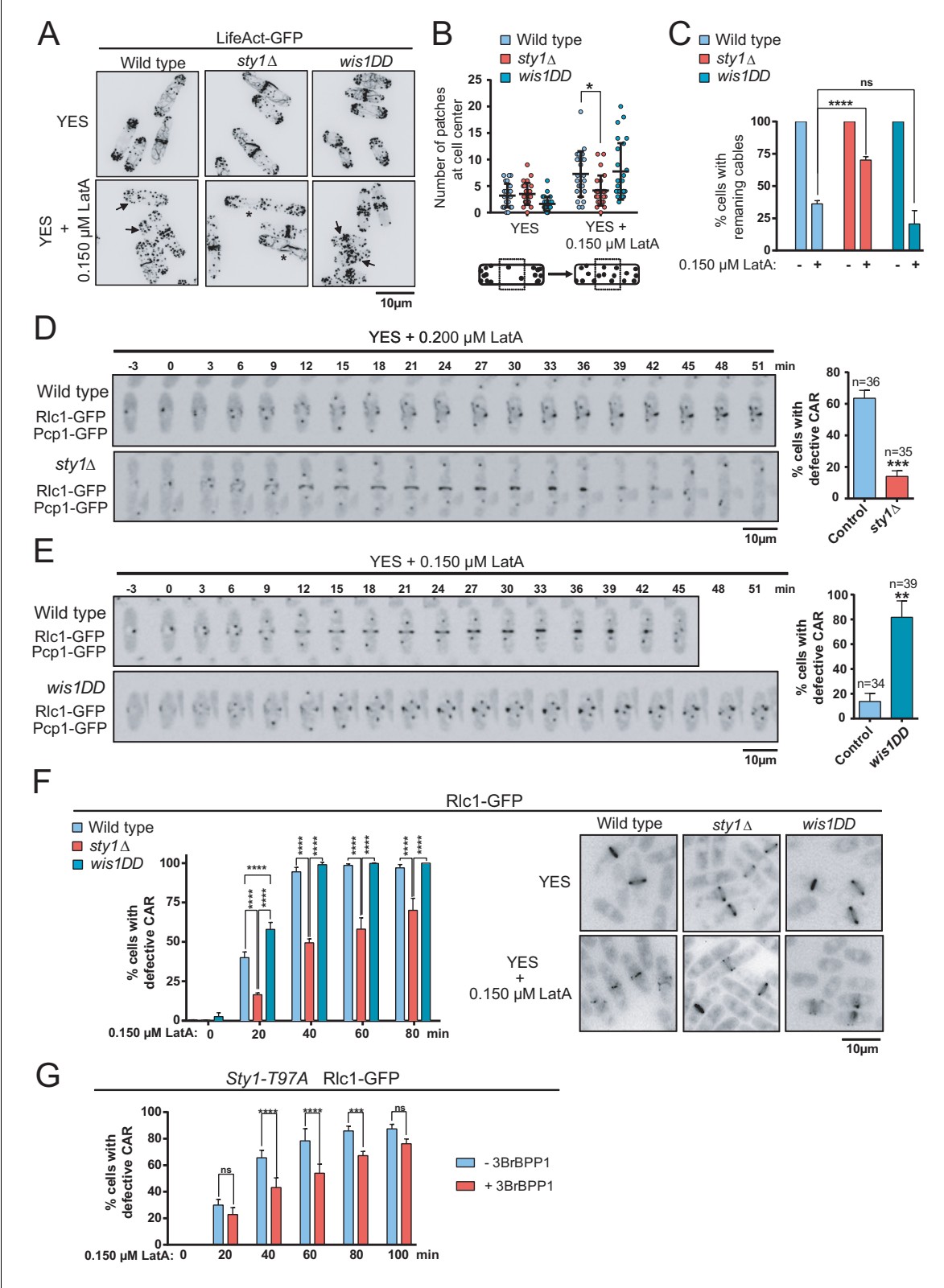

**Figure 2.** The SAPK pathway negatively regulates CAR assembly in response to cytoskeletal perturbations induced with LatA. (**A**) Representative maximum-projection images of *S. pombe* wild type, *sty1Δ* and *wis1DD* cells expressing the F-actin marker LifeAct-GFP growing in YES medium to mid-log phase, and remained untreated (0), or treated with 0.15 µM LatA for 30 min. Arrows indicate cells with depolarized actin patches, whereas asterisks show cells with persistent actin cables. (**B**) The number of actin patches at the medial region of interphasic G2 cells (n = 25) expressing the F-actin

*Figure 2 continued on next page*

*Figure 2 continued*

marker LifeAct-GFP was quantified in wild type, *sty1Δ* and *wis1DD* cultures described in (**A**) with or without 0.15 μM LatA treatment (30 min), and is represented as a dot plot with mean ± SD. *, p<0.05, as calculated by unpaired Student's *t* test. (**C**) The percentage of cells with actin cables (n > 100) was quantified in wild-type, *sty1Δ* and *wis1DD* cultures described in (**A**) during unperturbed growth (-) and after 0.15 μM LatA treatment for 30 min (+), and is represented as mean ± SD (biological triplicates). ****, p<0.0001; ns, not significant, as calculated by unpaired Student's *t* test. (**D**) Representative time-lapse maximum-projection images of Rlc1-GFP dynamics at the equatorial region of *S. pombe* wild-type and *sty1Δ* cells growing in YES medium to mid-log phase in the presence of 0.2 μM LatA. Mitotic progression was monitored using Pcp1-GFP-marked SPBs. Time interval is 3 min. Right: the percentage of cells that show damaged/defective CAR was estimated for each strain (n> 35) and is presented as mean ± SD (biological triplicates). ***, p<0.001, as calculated by unpaired Student's *t* test. (**E**) Representative time-lapse maximum-projection images of Rlc1-GFP dynamics at the equatorial region of *S. pombe* wild type and *wis1DD* cells growing in YES medium to mid-log phase in the presence of 0.15 μM LatA. Mitotic progression was monitored using Pcp1-GFP-marked SPBs. Time interval is 3 min. Right: the percentage of cells that show damaged/defective CAR was estimated for each strain (n> 34) and is presented as mean ± SD (biological triplicates). **, p<0.01, as calculated by unpaired Student's *t* test. (**F**) *S. pombe* wild type, *sty1Δ* and *wis1DD* cells expressing a Rlc1-GFP genomic fusion were grown in YES medium to mid-log phase, and remained untreated (0) or treated with 0.15 μM LatA for the indicated times. Left panel: the percentage of cells that show damaged CAR was estimated in each case by fluorescence microscopy (n > 200), and is presented as mean ± SD (biological triplicates). ****, p<0.0001; ns, not significant, as calculated by unpaired Student's *t* test. Right panel: representative images of cells from the above strains growing in YES medium either untreated or after treatment with 0.15 μM LatA for 40 min. (**G**) *S. pombe* Sty1-T97A (analogue sensitive) cells expressing a Rlc1-GFP genomic fusion were grown in YES medium to early log-phase, incubated with 10 μM 3BrBPP1 or with the solvent alone (methanol) for 8 hr, and then treated with 0.15 μM LatA during the indicated times. The percentage of cells that show damaged CAR was estimated in each case by fluorescence microscopy (n > 200), and is presented as mean ± SD (biological triplicates). ****, p<0.0001; ***, p<0.001; ns, not significant, as calculated by unpaired Student's *t* test.

The online version of this article includes the following source data and figure supplement(s) for figure 2:

**Source data 1.** Values used for graphical representations and statistical analysis in *Figure 2*.

**Figure supplement 1.** *act1⁺* mRNA levels in SAPK mutants.

**Figure supplement 2.** Act1 protein levels in SAPK mutants.

**Figure supplement 3.** LatA-induced depolymerization of F-actin patches in wild-type versus *sty1Δ* cells.

**Figure supplement 4.** *S. pombe* wild-type and *sty1Δ* strains expressing the F-actin marker LifeAct-GFP were grown in YES medium to mid-log phase and either remained untreated, incubated at 40°C, or treated with 0.6 M KCl or 1 mM H₂O₂ for 30 min.

**Figure supplement 5.** CAR dynamics during unperturbed growth in SAPK and For3 mutants.

(*Figure 2A and C*). Actin cables also disappeared quickly in *wis1DD* cells, but remained assembled in *sty1Δ* cells when subjected to the above conditions (*Figure 2A and C*). Moreover, the percentage of cells with remaining cables became reduced in wild-type cells in response to environmental stimuli that activate Sty1, like osmotic saline stress (0.6 M KCl), heat shock (40°C), or oxidative stress (1 mM H₂O₂) (*Pérez and Cansado, 2010*), but not in *sty1Δ* cells (*Figure 2—figure supplement 4*). These results suggest that Sty1 negatively regulates actin cable integrity in response to environmental stimuli and actin perturbations induced with LatA.

The dynamics of CAR assembly and constriction was assessed by employing time-lapse live fluorescence microscopy in wild-type, *sty1Δ*, and *wis1DD* cells (constitutive hyperactivated Sty1) cells co-expressing genomic C-terminal tagged GFP fusions of the myosin II regulatory light chain contractile ring component Rlc1 and Pcp1 (pericentrin SPB component as an internal control for mitotic progression). Despite the apparent differences in the cell length at division, the times for CAR assembly, maturation and constriction were not significantly altered in *sty1Δ* (elongated cells) and *wis1DD* mutants (shorter cells) during unperturbed growth with respect to wild-type cells (*Figure 2—figure supplement 5*). Treatment with 0.2 μM LatA that inhibits growth in solid rich medium (*Figure 1B*), impaired mature CAR assembly from cytokinetic nodes in >80% of wild-type cells growing in liquid medium, resulting in the formation of many disorganized filaments that remained over time, whereas rings formed and constricted correctly in >85% of *sty1Δ* cells (*Figure 2D*). In the presence of 0.15 μM LatA rings formed and constricted normally in >85% of wild type cells, whereas in >84% of *wis1DD* cells cytokinetic nodes failed to coalesce into a compact and functional ring (*Figure 2E*).

Remarkably, in asynchronous liquid cultures ~ 40% of wild-type cells expressing Rlc1-GFP displayed defective rings after 20 min of treatment with 0.15 μM LatA, and this percentage raised to ~90% after 40 min of treatment (*Figure 2F*). Again, *wis1DD* displayed a more pronounced CAR damage than wild-type cells (55% at 20 min), while CAR integrity was maintained for more extended periods in LatA-treated *sty1Δ* cells (*Figure 2F*). Moreover, node coalescence for CAR assembly was favored in LatA-treated Rlc1-GFP cells expressing an analogue-sensitive version of the MAPK (Sty1-

T97A) (*Tournier et al., 2004*) after 8 hr of incubation in the presence of the kinase specific inhibitor (10 µM 3BrBPP1 in methanol), with respect to those treated with solvent alone (*Figure 2G*). The higher percentages of cells with damaged CAR found during LatA treatment in asynchronous cultures as compared to the time-lapse experiments may be due to a combination of factors, including genotypic differences among strains, soaking up of small LatA concentrations by the chambers during time-lapse observations, and the different scoring method, since in asynchronous cultures cells with seemingly damaged CARs might still be able to assemble and constrict. In any case, the above results suggest that in *S. pombe* the SAPK pathway negatively regulates CAR assembly and maturation in response to cytoskeletal perturbations induced with LatA.

## For3 function is essential for SAPK-dependent negative regulation of ring assembly in response to actin cytoskeleton damage

For3, one of the three formins present in *S. pombe*, assembles actin filaments within long cables for cellular transport and has established roles in cell polarity (*Feierbach and Chang, 2001*; *Martin and Chang, 2006*; *Martin et al., 2007*; *Scott et al., 2011*). For3 also cooperates with the essential formin Cdc12 in cytokinesis during contractile ring assembly and disassembly (*Coffman et al., 2013*). The time spent for node condensation and ring maturation in *for3Δ* cells during unperturbed growth was very similar to that of wild-type cells. However, and confirming previous observations (*Coffman et al., 2013*), they displayed an explicit delay in ring constriction and disassembly (21 ± 0.6 min in wild-type cells *vs* 36 ± 1.6 min in *for3Δ* cells; *Figure 2—figure supplement 5*). These precedents, together with the observation that actin cables remained mostly assembled in *sty1Δ* cells in response to LatA or to other environmental stress (*Figure 2C* and *Figure 2—figure supplement 4*, respectively), suggested that negative regulation of CAR integrity by the SAPK pathway might involve For3 function.

*for3Δ* cells showed a clear growth-sensitive phenotype in the presence of LatA as compared to wild type cells (*Figure 3A*). Moreover, For3 deletion completely suppressed the growth-resistant phenotype of *sty1Δ* cells in the presence of LatA (*Figure 3A*). Timelapse microscopy showed that CARs failed to properly assemble in >70% of *for3Δ* cells in the presence of 0.15 µM LatA, whereas they formed and constricted accurately in >80% of wild-type cells under the same conditions (*Figure 3B*). These results suggest that For3 localization at the CAR is essential to maintain ring integrity and support cell growth in response to LatA. CAR integrity during LatA treatment could not be assessed in a double *sty1Δ for3Δ* mutant expressing the Rlc1-GFP fusion since it shows a synthetic lethal phenotype. Similarly, *sty1Δ for3Δ* mutants expressing other GFP-fused CAR components, including Myo2-GFP or Cdc15-GFP were not viable. Hence, quantification of CAR integrity in the presence of LatA was determined in wild-type, *sty1Δ*, *for3Δ*, and *sty1Δ for3Δ* fixed cells by immunofluorescence microscopy after staining with AlexaFluor488-phalloidin. Both wild-type and *for3Δ* cells showed highly damaged rings when incubated for 40 min in the presence of 0.15 µM LatA, whereas ~45% of rings remained unaffected in *sty1Δ* cells (*Figure 3C*). However, simultaneous deletion of For3 in *sty1Δ* cells (*sty1Δ for3Δ* double mutant) elicited defects in CAR integrity similar to those of *for3Δ* cells (*Figure 3C*).

For3 cannot localize and bind to cortical tethers when present in a closed, autoinhibited state, which is mediated, similar to other diaphanous-like formins, by an intramolecular interaction between its functional DAD and DID domains (*Martin et al., 2007*). Cells expressing a constitutively open GFP-fused genomic version of For3 (For3(DAD)−2GFP) (*Martin et al., 2007*), showed more resistance to LatA than those expressing a wild-type formin (For3-3GFP) (*Figure 4A*). Hence, treatment of exponentially growing liquid cultures with 0.2 µM LatA induced a clear CAR assembly defect in ~75% of For3-3GFP cells, but not in those expressing For3(DAD)−2GFP (~12%) (*Figure 4B*). Additionally, For3(DAD)−2GFP expression significantly alleviated the CAR integrity defects of *wis1DD* cells in the presence of 0.15 µM LatA (*Figure 4C*).

The SAPK pathway might also negatively regulate the Cdc12 function, which is essential for assembly of actin filaments to the CAR (*Pollard and Wu, 2010*; *Chang et al., 1997*; *Kovar et al., 2003*), and is tightly regulated by Sid2, the terminal kinase in the SIN (*Bohnert et al., 2013*), and the mitotic kinase Cdk1 (Cdc2) (*Willet et al., 2018*). Phosphorylation of a genomic Cdc12-3HA fusion in wild-type cells (PhosTag) does slightly increase as the concentration of LatA increases, whereas total formin levels remained constant (*Figure 3—figure supplement 1*). However, Cdc12 protein and phosphorylation status did not change in LatA-treated *sty1Δ* or *wis1DD* mutants as

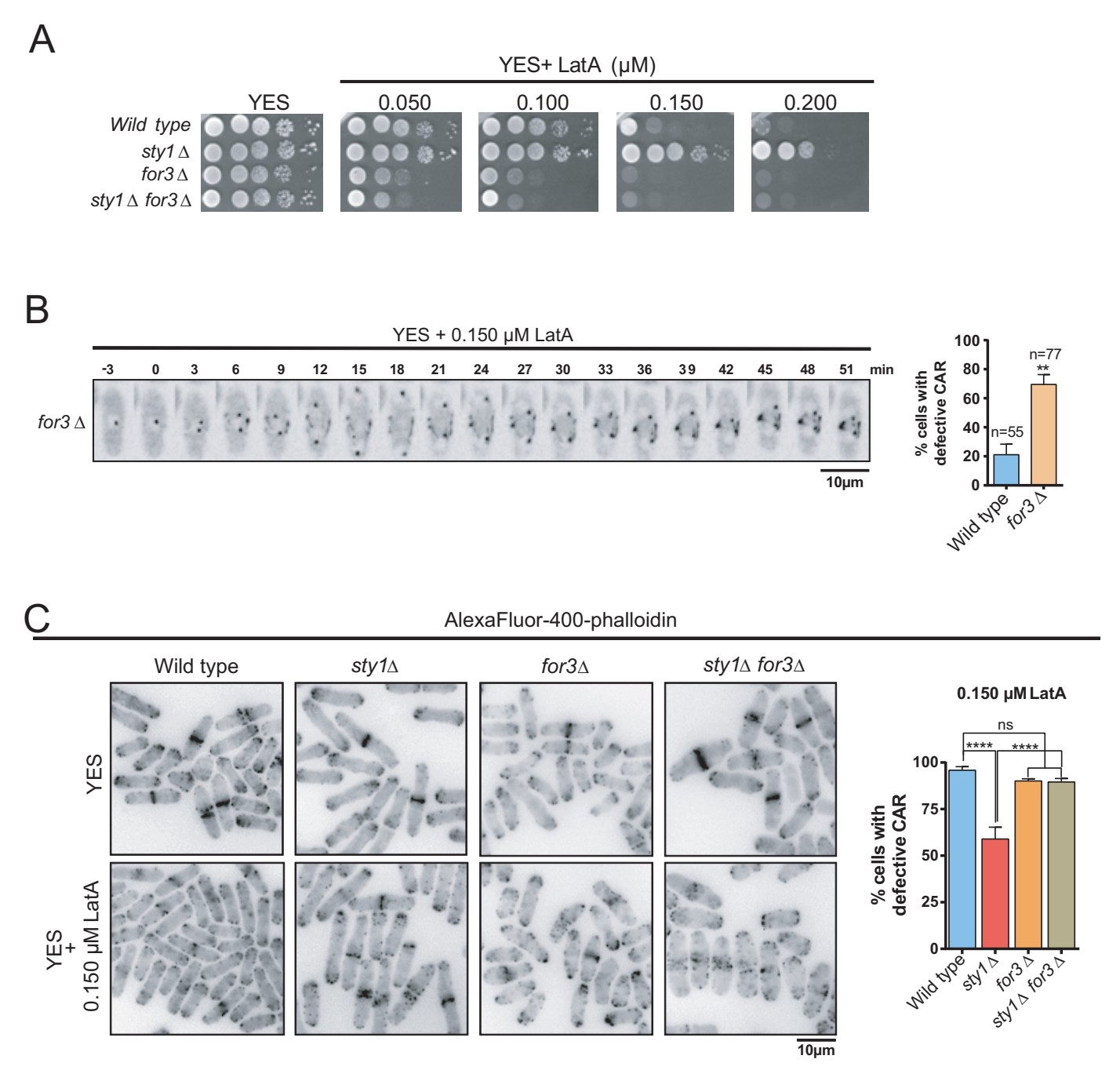

**Figure 3.** SAPK-mediated negative control of CAR assembly in response to actin cytoskeleton damage relies on formin For3. (**A**) Decimal dilutions of strains of the indicated genotypes were spotted on YES and YES solid plates with a range of concentrations of LatA, incubated at 30°C for 3 days, and photographed. A representative experiment is shown. (**B**) Representative time-lapse maximum-projection images of Rlc1-GFP dynamics at the equatorial region of *S. pombe for3Δ* cells growing in YES medium to mid-log phase in the presence of 0.15 µM LatA. Mitotic progression was monitored using Pcp1-GFP-marked SPBs. Time interval is 3 min. Right: the percentage of cells that show damaged/defective CAR was estimated both for wild-type and *for3Δ* cells (n> 55), and is presented as mean ± SD (biological triplicates). **, p<0.01, as calculated by unpaired Student's *t* test. (**C**) Left panel: representative images of Alexa Fluor–phalloidin staining of *S. pombe* wild-type, *sty1Δ*, *for3Δ*, and *sty1Δ for3Δ* cells growing in YES medium to mid-log phase, and remained untreated, or treated with 0.15 µM LatA for 40 min. Right panel: the percentage of cells that show damaged/defective CAR was estimated for each strain (n> 200) and is presented as mean ± SD (biological triplicates). ****, p<0.0001; ns, not significant, as calculated by unpaired Student's *t* test.

The online version of this article includes the following source data and figure supplement(s) for figure 3:

*Figure 3 continued on next page*

*Figure 3 continued*

**Source data 1.** Values used for graphical representations and statistical analysis in *Figure 3*.
**Figure supplement 1.** Total levels and mobility of formin Cdc12 in response to a range of concentrations of LatA.
**Figure supplement 2.** Total levels and mobility of formin Cdc12 in SAPK mutants treated with LatA.
**Figure supplement 3.** Lack of SAPK activity alleviates LatA-sensitivity of Cdc12 phospho-mutants.
**Figure supplement 4.** Lack of SAPK activity alleviates the CAR assembly defects of Cdc12 phospho-mutants induced with LatA.

compared to wild-type cells (*Figure 3—figure supplement 2*). Moreover, Sty1 deletion partially suppressed the LatA-sensitivity and CAR defects of a *cdc12-4A* strain in which Sid2 phospho-sites are mutated to alanines (*Bohnert et al., 2013*; *Figure 3—figure supplement 3*), and a Cdc12 mutant with Cdk1 sites changed to phosphomimetic aspartic acid residues (*cdc12-6D*) (*Willet et al., 2018*; *Figure 3—figure supplements 3* and *4*). These findings suggest that the negative effect of SAPK signaling in *S. pombe* CAR assembly mostly relies on For3 function.

## Sty1 activity negatively regulates protein levels of For3 in response to stress

The finding that the adverse effects of Sty1 signaling on the CAR depend on For3 prompted us to explore further how this control may be exerted. In exponentially growing wild-type cells, a genomic For3-3GFP fusion migrates in SDS-PAGE as three bands (*Figure 5A*). Lambda phosphatase treatment of cell extracts with or without specific phosphatase inhibitor revealed that the two bands with lower mobilities are phosphorylated (*Figure 5A*). For3-3GFP levels and phosphorylation status did not change significantly during cell cycle as evidenced by comparative western blot analysis of cells arrested at G1 (*cdc10-129*), G2 (*cdc25-22*), and M (*nda3-KM311*) phases (*Figure 5—figure supplement 1*). Interestingly, as compared to wild-type cells, total For3-3GFP levels were significantly raised in *sty1Δ* cells (~1.7 times), and lower in *wis1DD* and *pyp1Δ* cells that have higher Sty1 basal activity (*Figure 5B*; *Madrid et al., 2007*). Moreover, in a strain expressing For3-3GFP and an analogue-sensitive version of the MAPK (Sty1-T97A) (*Prieto-Ruiz et al., 2020*), total For3-3GFP levels increased progressively after 2–3 hr in the presence of 10 µM 3BrBPP1 kinase-specific inhibitor, while its phosphorylation pattern remained unchanged (*Figure 5C*). This observation rules out the possibility that Sty1 deletion might lead to a long-term physiological adaptation that alters For3 status in the absence of stress. Higher For3-3GFP levels in the absence of Sty1 function resulted in an overall enhanced cellular localization of the formin including the CAR, as confirmed by comparative quantitative fluorescence microscopy of mixed wild type and *sty1Δ* cells expressing For3-3GFP, with wild-type cells expressing Hht1-mCherry for easy discrimination between both strains (*Figure 5D*). Increased localization of For3 is not a general feature associated to Sty1 deletion, since localization at the CAR of the cell integrity pathway MAPK Pmk1-GFP fusion (*Madrid et al., 2006*) decreased slightly in *sty1Δ* cells as compared to wild-type cells (*Figure 5D*).

We also measured by qPCR analysis mRNA expression levels of *for3*[+]gene that, in contrast to protein levels, were reduced to approximately ~50% and~25% in *sty1Δ* cells and *atf1Δ* cells, respectively, and remained unchanged in *wis1DD* cells as compared to wild-type cells (*Figure 5—figure supplement 2*). Additionally, in wild-type cells *for3*[+] mRNA expression increased gradually in response to a range of LatA concentrations (*Figure 5E*). Notably, these conditions, which induce a progressive rise in Sty1 phosphorylation, prompted a concomitant decrease in For3-3GFP protein levels instead (*Figure 5F*). We did not observe a significant decrease of For3-3GFP levels at early incubation times (0–60 min) in wild-type cells treated with low LatA concentrations (0.150–0.200 µM), even though a high percentage of cells (~80%) showed damaged CAR under these conditions (*Figure 2G*). Perhaps, this decline might be subtle and below the limit of detection of the western blot assay. Alternatively, specific pools of the formin may undergo subtle reductions at defined subcellular locations like the CAR. In agreement with this prediction, For3 levels decreased quickly (30 min) in synchronized *cdc2-asM17* For3-3GFP cells treated with 0.200 µM LatA 10 min after release from the G2 arrest (*Figure 5G*), as they are entering mitosis (~20% binucleated cells) and undergoing CAR assembly. This observation suggests that reduction of For3 levels is an essential factor responsible for the CAR integrity defects in the presence of LatA.

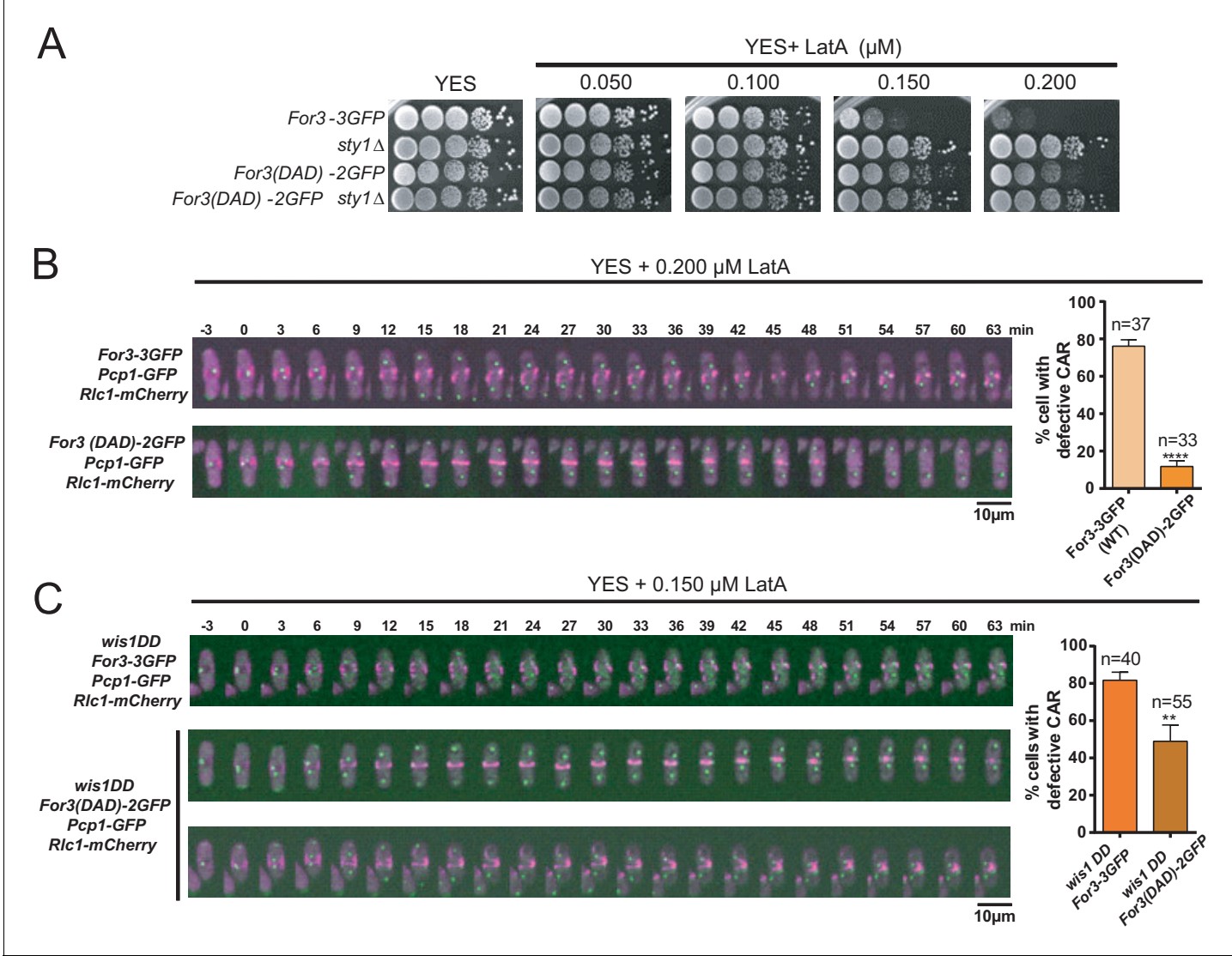

**Figure 4.** A constitutively open version of For3 favors growth and CAR integrity in the presence of LatA. (**A**) Decimal dilutions of strains of the indicated genotypes were spotted on YES and YES solid plates with a range of concentrations of LatA, incubated at 30°C for 3 days, and photographed. A representative experiment is shown. (**B**) Representative time-lapse maximum-projection images of Rlc1-mCherry dynamics at the equatorial region of *S. pombe* For3-3GFP and For3(DAD)−2GFP cells growing in YES medium to mid-log phase in the presence of 0.2 μM LatA. Mitotic progression was monitored using Pcp1-GFP-marked SPBs. Time interval is 3 min. Right: the percentage of cells that show damaged/defective CAR was estimated for each strain (n> 33) and is presented as mean ± SD (biological triplicates). ****, p<0.0001, as calculated by unpaired Student's *t* test. (**C**) Representative time-lapse maximum-projection images of Rlc1-mCherry dynamics at the equatorial region of *S. pombe* wis1DD For3-3GFP and *wis1DD* For3(DAD)−2GFP cells growing in YES medium to mid-log phase in the presence of 0.2 μM LatA. Mitotic progression was monitored using Pcp1-GFP-marked SPBs. Time interval is 3 min. Right: the percentage of cells that show damaged/defective CAR was estimated for each strain (n> 39) and is presented as mean ± SD (biological triplicates). **, p<0.01, as calculated by unpaired Student´s *t* test.

The online version of this article includes the following source data for figure 4:

**Source data 1.** Values used for graphical representations and statistical analysis in *Figure 4*.

---

Reduction in formin levels was absent in cells expressing the LatA-resistant actin mutant allele *Act1-R183A D184A* (*Figure 5H*), which does not activate Sty1 (see *Figure 1—figure supplement 2*), implying that this outcome is a consequence of MAPK activation during actin network/s damage. We found that total For3 levels also decline in *S. pombe* wild-type cells in response to stimuli that activate Sty1, like heat shock (40°C), osmotic saline (0.6 M KCl), and oxidative stress (1 mM $H_2O_2$) (*Pérez and Cansado, 2010*) in a MAPK-dependent manner (*Figure 5—figure supplement 3*). Importantly, in contrast to the wild-type, *wis1DD*, and *pyp1Δ* cells, For3-3GFP levels remained unchanged

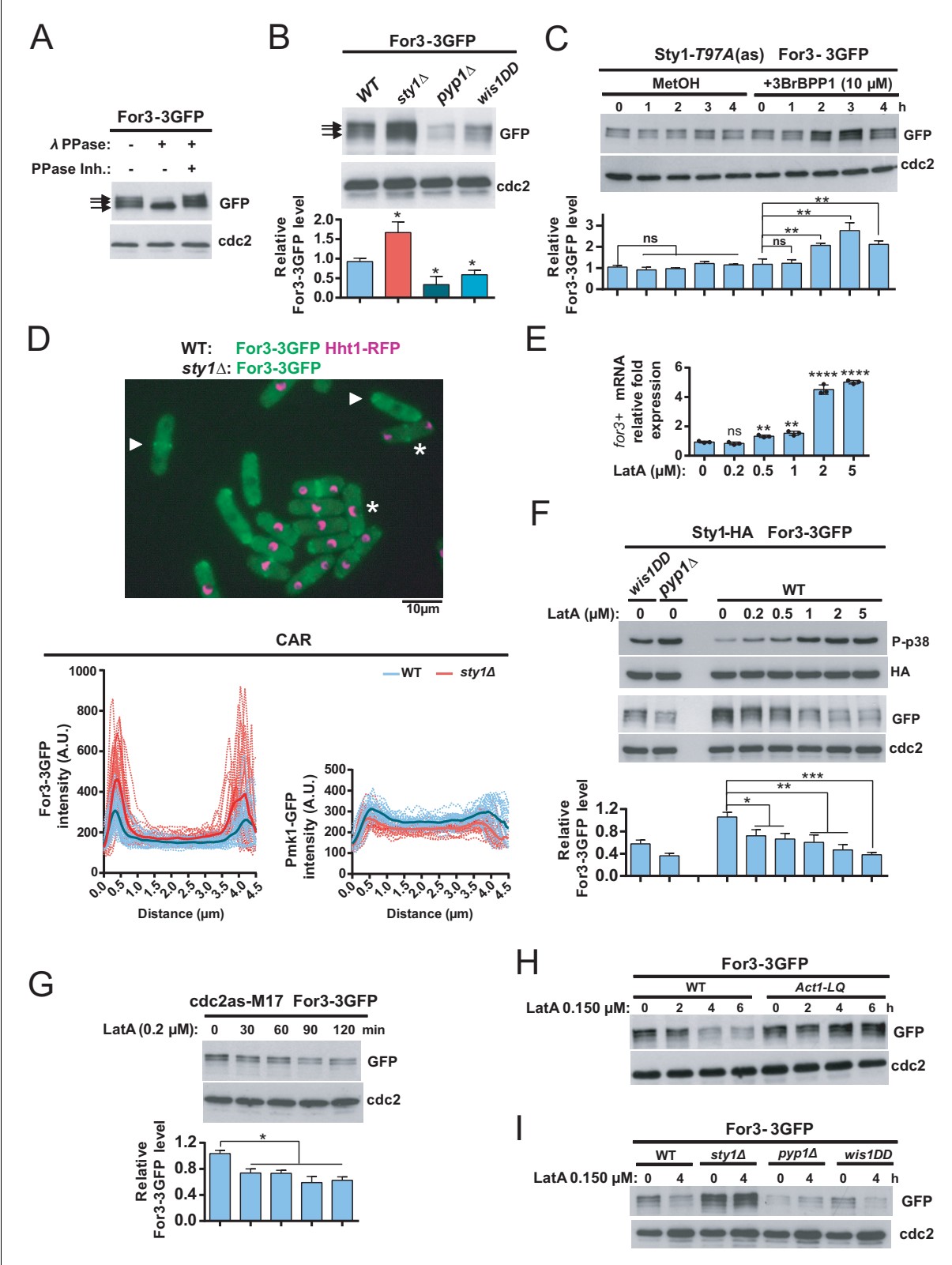

**Figure 5.** Sty1 MAPK prompts a decrease in total protein levels of For3 during actin cytoskeleton damage induced with LatA. (**A**) Extracts from *S. pombe* growing cells expressing a genomic For3-3GFP fusion were treated with lambda phosphatase in the presence/absence of specific phosphatase inhibitor. Total and phosphorylated For3 levels (arrows) were determined by immunoblotting with anti-GFP antibody. Anti-Cdc2 was used as a loading control. Results from a representative experiment are shown. (**B**) Total extracts from growing cultures of wild-type, *sty1Δ*, *pyp1Δ*, and *wis1DD* strains

*Figure 5 continued on next page*

Figure 5 continued

expressing a For3-3GFP genomic fusion were resolved by SDS-PAGE, and total For3 levels were detected by incubation with anti-GFP antibody. Anti-Cdc2 was used as a loading control. Lower panel: quantification of western blot experiments. Relative For3 levels are represented as mean ± SD (biological triplicates). *, p<0.05, as calculated by unpaired Student's *t* test with respect to the wild type. (C) Sty1-T97A (as) cells expressing a For3-3GFP genomic fusion were grown to mid-log phase and remained untreated (MetOH; solvent control), or treated with 10 µM 3BrBPP1 at the indicated times. Total extracts were resolved by SDS-PAGE, and For3 levels were detected by incubation with anti-GFP antibody. Anti-Cdc2 was used as a loading control. Lower panel: quantification of western blot experiments. Relative For3 levels are represented as mean ± SD (biological duplicates). **, p<0.005; ns, not significant, as calculated by unpaired Student's *t* test. (D) Upper panel: Representative maximum-projection image of mixed wild type (For3-3GFP, Hht1-RFP; representative cells are marked with asterisks) and *sty1Δ* (For3-3GFP; representative cells are marked with point arrows) cells growing to mid-log phase observed by fluorescence microscopy. Lower panels: intensity plots of For3-3GFP and Pmk1-GFP fusions (shown as arbitrary fluorescence units) were generated from line scans across the equatorial region of both wild-type and *sty1Δ* cells (n> 28) with early septum. Individual (dotted lines) and average scans (solid lines) are shown in each case. (E) mRNA levels of *for3+* gene were measured by qPCR from total RNA extracted from cell samples corresponding to *S. pombe* wild type cells growing exponentially in YES medium that remained untreated (0), or treated with the indicated concentrations of LatA for 1 hr. Results are shown as relative fold expression (mean ± SD) from three biological repeats. ****, p<0.0001; **, p<0.005; ns, not significant, as calculated by unpaired Student's *t* test with respect to the wild type. (F) *S. pombe* wild-type, *wis1DD* and *pyp1Δ* cells expressing genomic Sty1-HA6his and For3-3GFP fusions were grown in YES medium to mid-log phase, and remained untreated (0), or treated with the indicated concentrations of LatA for 2 hr. Activated/total Sty1 were detected with anti-phospho-p38 and anti-HA antibodies, respectively. Total For3 levels were detected with anti-GFP antibody. Anti-Cdc2 was used as a loading control. Lower panel: quantification of Western blot experiments. Relative For3 levels are represented as mean ± SD (biological duplicates). ***, p<0.001; **, p<0.005; *, p<0.05, as calculated by unpaired Student's *t* test. (G) Exponentially growing *cdc2-asM17* cells expressing a For3-3GFP fusion were treated with 1 µM 3-MB-PP1 for 3 hr to hold the cycle at G2, released from the arrest for 10 min after ATP-analogue washout, and treated with 0.2 µM LatA for the indicated times. Total For3 levels were detected with anti-GFP antibody. Anti-Cdc2 was used as a loading control. Lower panel: quantification of Western blot experiments. Relative For3 levels are represented as mean ± SD (biological duplicates). *, p<0.05, as calculated by unpaired Student's *t* test. (H) Wild type and *Act1-LQ* (LatA-insensitive mutant) cells expressing a For3-3GFP genomic fusion were grown to mid-log phase and remained untreated (0), or treated with 0.15 µM LatA for 2, 4, or 6 hr. Total extracts were resolved by SDS-PAGE, and For3 levels were detected by incubation with anti-GFP antibody. Anti-Cdc2 was used as a loading control. Results from a representative experiment are shown. (I) Wild-type, *sty1Δ*, *pyp1Δ*, and *wis1DD* cells expressing a For3-3GFP genomic fusion were grown to mid-log phase and remained untreated (0), or treated with 0.15 µM LatA for 4 hr. Total extracts were resolved by SDS-PAGE, and For3 levels were detected by incubation with anti-GFP antibody. Anti-Cdc2 was used as a loading control. Results from a representative experiment are shown.

The online version of this article includes the following source data and figure supplement(s) for figure 5:

**Source data 1.** Values used for graphical representations and statistical analysis in *Figure 5*.
**Figure supplement 1.** For3-3GFP levels and phosphorylation status do not change during the cell cycle.
**Figure supplement 2.** *for3+* mRNA levels in SAPK mutants.
**Figure supplement 3.** Sty1 activity elicits a decrease in For3 levels in response to environmental stimuli.
**Figure supplement 4.** For3 is ubiquitinated in vivo.
**Figure supplement 5.** Half-life of For3 in wild type *versus sty1Δ* cells.
**Figure supplement 6.** Sty1 does not phosphorylate For3 in vitro.

in *sty1Δ* cells during LatA treatment or environmental stresses (*Figure 5I* and *Figure 5—figure supplement 3*). As a whole, these observations indicate that in fission yeast activation of the SAPK pathway and its effector Sty1 prompts a specific decrease in total levels of For3 protein that accounts for lower CAR stability. They also suggest that negative control of For3 by the SAPK pathway takes place mostly at the post-transcriptional level.

mDia2 formin, a key nucleator of actin filaments during cell migration in mammalian cells, is degraded at the end of mitosis through ubiquitination, and this modification is essential for the completion of cell division (*DeWard and Alberts, 2009*). By performing ubiquitin pull-down assays ubiquitinated For3-3GFP species were recovered from wild-type cells during unperturbed growth at 28°C (*Figure 5—figure supplement 4*). Moreover, as compared to wild type cells, For3 ubiquitination ratio was more pronounced in a temperature-sensitive mutant of the proteasome subunit Mts3/Rpn12 (*mts3-1* For3-3GFP cells) when incubated at the restrictive temperature (36°C) (*Figure 5—figure supplement 4*). Considering that the decrease in total For3 levels at high temperature is Sty1-dependent (*Figure 5—figure supplement 3*), these findings suggest that a SAPK-mediated mechanism enhances For3 in vivo ubiquitination and degradation in response to stress.

The actin-binding proteins profilin (Cdc3) and tropomyosin (Cdc8) localize to the contractile ring and play essential roles in the formation, stabilization, and maintenance of actin filaments for the CAR during cytokinesis (*Balasubramanian et al., 1992*; *Balasubramanian et al., 1994*). Cells

expressing the hypomorphic *cdc3-124* and *cdc8-110* alleles are defective in CAR assembly and colony formation at 32°C and 34°C, respectively (*Figure 6A*; *Balasubramanian et al., 1996*; *Zambon et al., 2020*). Incubation of *cdc3-124* and *cdc8-110* cells during 4 hr at the restrictive temperature (34°C), prompted a reduction in total protein levels of a genomic For3-3GFP fusion as compared to wild type cells (*Figure 6B*). Remarkably, Sty1 deletion partially alleviated the thermosensitivity of *cdc3-124* cells but not that of the *cdc8-110* allele (*Figure 6A*), and restored For3 levels in both mutants growing at 34°C (*Figure 6B*). Hence, a reduction in profilin or tropomyosin function decreases For3 levels in a Sty1-dependent fashion (i.e. in the absence of LatA treatment), further revealing the general significance of the SAPK-For3 branch during the control of actin cables and CAR integrity in response to stress.

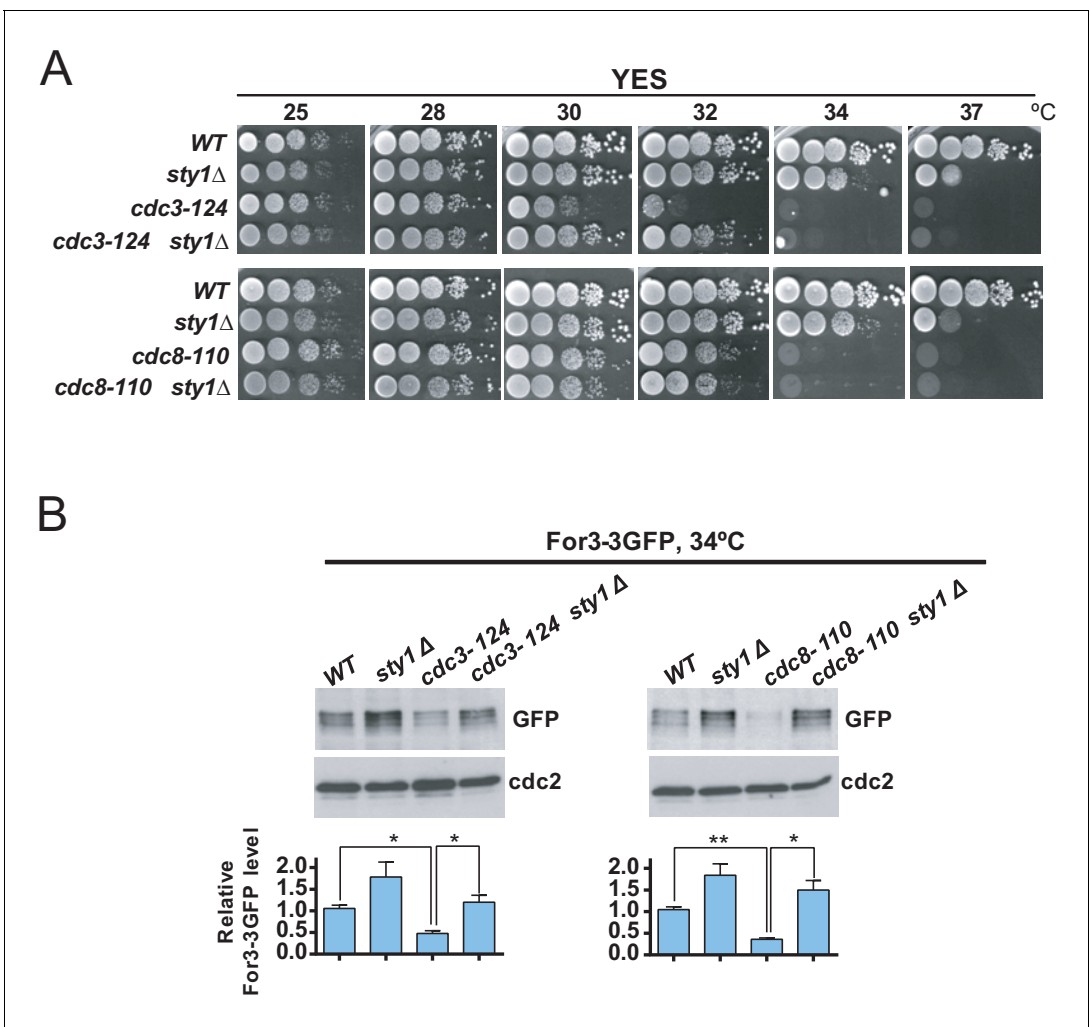

**Figure 6.** Decreased For3 levels in profilin and tropomyosin conditional mutants are relieved in absence of SAPK function. (**A**) Decimal dilutions of strains of the indicated genotypes were spotted on YES solid plates, incubated at 25, 28, 30, 32, 34, and 37°C for 3 days, and photographed. (**B**) Growing cultures of the indicated strains expressing a For3-3GFP genomic fusion were incubated at 34°C for 4 hr. Total extracts obtained from the respective cell aliquots were resolved by SDS-PAGE, and total For3 levels were detected by incubation with anti-GFP antibody. Anti-Cdc2 was used as a loading control. Lower panel: quantification of western blot experiments. Relative For3 levels are represented as mean ± SD (biological duplicates). **, $p < 0.005$; *, $p < 0.05$, as calculated by unpaired Student's *t* test.

The online version of this article includes the following source data for figure 6:

**Source data 1.** Values used for graphical representations and statistical analysis in *Figure 6*.

## For3 is a dose-dependent regulator of CAR integrity in *S. pombe*

If the SAPK pathway downregulates For3 levels to negative control CAR integrity, then inducing progressive ectopic expression of the formin should diminish CAR damage in the presence of LatA. To test this hypothesis, we employed the β-estradiol-regulated system based on the Z₃EV synthetic transcription factor recently implemented for *S. pombe* (*Ohira et al., 2017*) and constructed a *for3Δ* strain in which the expression of a For3-HA C-terminal fusion is under control of the Z₃EV promoter. As shown in *Figure 7A*, total levels of a genomic For3-HA fusion were similar to those present after incubation of *for3Δ* Z3EV:For3-HA cells for 3 hr in the presence of 10 nM β-estradiol, and increased approximately 3–4 fold after 6 hr of incubation. As compared to cells expressing wild type-like levels of the formin, enhanced For3 levels led to a significant reduction in CAR damage during treatment with 0.15 μM LatA, while strong CAR defects were observed 12 hr after β-estradiol removal again (*Figure 7B*). Therefore, For3 behaved as a dose-dependent regulator of CAR integrity in *S. pombe*.

## The SAPK pathway promotes CAR integrity in *S. japonicus*

The SAPK pathway functions, including cell cycle control, adaptation to stress conditions, and regulation of sexual differentiation, are evolutionarily conserved both in *S. pombe* and *Schizosaccharomyces japonicus*. However, unlike *S. pombe*, *S. japonicus* delays the assembly of a centrally positioned ring until chromosome segregation is complete (*Gomez-Gil et al., 2019*). Hence, we wondered if the SAPK-dependent negative control of CAR integrity present in *S. pombe* is also shared by *S. japonicus*. We found that *S. japonicus* LatA-sensitivity on solid medium was approximately one order of magnitude higher than in *S. pombe* (*Figure 8A*). By employing phospho-p38 antibodies (*Gomez-Gil et al., 2019*), we observed that, similar to *S. pombe*, *S. japonicus* Sty1 became increasingly activated after 1 hr of treatment with LatA in a dose-dependent manner (*Figure 8B*). However, in sharp contrast to *S. pombe* SAPK mutants, *S. japonicus* *sty1Δ* and *atf1Δ* cells were strongly hypersensitive to LatA (*Figure 8A*). Accordingly, time-lapse fluorescence microscopy of exponentially growing *S. japonicus* strains co-expressing Rlc1-GFP and Pcp1-mCherry fusions and treated with 0.005 μM LatA revealed that, in contrast to wild-type cells, most cytokinetic nodes in *sty1Δ* cells (~90%) were unable to coalesce and form a compact and functional CAR (*Figure 8C*). These results suggest that *S. japonicus* SAPK pathway may have a positive impact on CAR assembly and integrity during stress induced with LatA.

## Discussion

Among its numerous essential functions, MAPKs participate in the dynamic remodeling of the actin cytoskeleton in eukaryotic organisms, and generally play a positive role reinforcing the actin-based structures, including the CAR (*Pullikuth and Catling, 2007*). However, as opposed to this shared vision, we show in this work that fission yeast activation of the SAPK Sty1, a p38 MAPK ortholog, negatively regulates CAR assembly and integrity in response to actin polymerization inhibition induced with LatA. We found that Sty1 becomes activated by low concentrations of this drug that specifically depolymerize actin cables and the CAR in wild-type cells. Strikingly, mutants lacking Sty1 or upstream activators of the SAPK pathway were more resistant to LatA than wild-type cells, and Sty1-hyperactive *wis1DD* cells failed to condense functional rings in the presence of LatA concentrations that were permissive for CAR assembly in wild-type cells.

Of the two *S. pombe* formins that function in the vegetative cycle, Cdc12 seemed a suitable candidate target for SAPK control of CAR integrity considering its essential role in the nucleation of actin filaments for ring assembly (*Chang et al., 1997*; *Kovar et al., 2003*). However, several evidences, which include comparative analysis of Cdc12 levels and phosphorylation status with LatA, plus growth assays of kinase-specific phospho-mutants with lack or gain of function in Sty1 signaling, suggested that the SAPK pathway does not target this formin. The other *S. pombe* vegetative formin, For3, is required for the assembly of actin cables that participate in polarized secretion and growth (*Feierbach and Chang, 2001*; *Martin and Chang, 2006*; *Martin et al., 2007*; *Scott et al., 2011*). Additionally, For3 function becomes essential in genetic backgrounds with severely impaired Cdc12 activity, suggesting that it cooperates with Cdc12 to nucleate actin filaments for CAR assembly (*Coffman et al., 2013*). We found that SAPK (Sty1) negative control

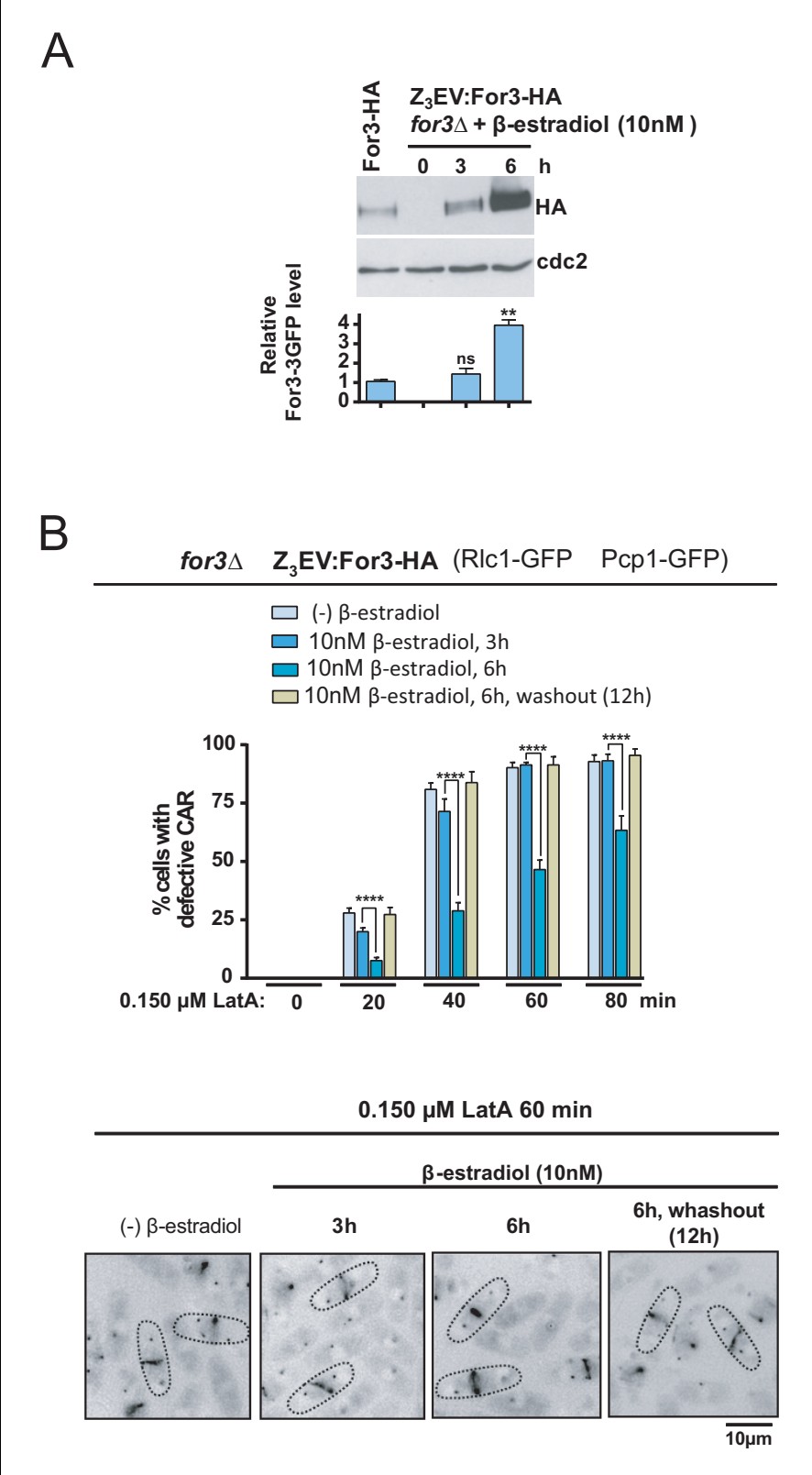

**Figure 7.** For3 is as a dose-dependent regulator of CAR assembly and integrity in *S. pombe*. (**A**) For3-HA (wild type) and *for3Δ* Z3EVpr:For3-HA strains were grown to mid-log phase, and the later culture was treated with 10 nM β-estradiol for 0, 3 and 6 hr. Total extracts were resolved by SDS-PAGE, and For3 levels were detected by incubation with anti-HA-HRP antibody. Anti-Cdc2 was used as a loading control. Relative For3 levels are

*Figure 7 continued on next page*

*Figure 7 continued*

represented as mean ± SD (biological duplicates). \*\*, p<0.005; ns, not significant, as calculated by unpaired Student's *t* test with respect to the wild type. (**B**) *for3Δ* Z3EVpr:For3-HA cells grown to mid-log phase were treated with 0.15 µM LatA for the indicated times after being incubated in absence (-) or presence of 10 nM β-estradiol for 3 and 6 hr, or 6 hr with β-estradiol plus 12 hr after hormone washout. Upper panel: the percentage of cells that show damaged CAR was estimated in each case by fluorescence microscopy (n > 200), and is presented as mean ± SD (biological triplicates). \*\*\*\*, p<0.0001, as calculated by unpaired Student's *t* test. Lower panel: representative images of cell samples after treatment with 0.15 µM LatA for 60 min are shown.

The online version of this article includes the following source data for figure 7:

**Source data 1.** Values used for graphical representations and statistical analysis in *Figure 7*.

---

of CAR integrity in response to stress is mainly exerted through For3, since deletion of this formin completely suppressed the enhanced CAR assembly and growth of *sty1Δ* cells with LatA. In vitro studies have shown that, as compared to Cdc12, For3 is a very poor nucleator of actin filaments (*Scott et al., 2011*), which suggests that sequestering of actin monomers induced by LatA might impair For3 nucleation activity more severely. Moreover, For3 inhibits actin assembly at low concentrations, while stimulating filaments assembly only at higher concentrations (*Scott et al., 2011*). Thus, from a biological perspective, precise control of For3 levels by the SAPK pathway might represent a suitable regulatory mechanism to fine tune its actin nucleation ability depending on the environmental conditions.

SAPK signaling elicits essential adaptive responses to ensure cell survival in response to environmental cues, like changes in osmolarity, redox status, temperature, or nutrient availability (*Pérez and Cansado, 2010*). Studies mostly performed in *S. cerevisiae* have shown that in response to the above stimuli, the actin cytoskeleton is targeted by signaling pathways including PKA, TOR and MAPKs, and that cellular adaptation to environmental changes often leads to its disassembly and remodeling (*Smethurst et al., 2014*). Thus, in *S. pombe* SAPK-mediated downregulation of CAR integrity might become biologically relevant when considered as part of a multifaceted adaptive response to environmental stress. This is sustained by our finding that reduction of For3 levels in response to stress is Sty1-dependent (*Figure 5—figure supplement 3*). In this context, recent work has shown that the presence of 0.6 M KCl does not prevent de novo formation of Myosin-II (Myo2) rings in *S. pombe* wild-type cells, but decreases the rate of constriction of the newly formed rings (*Okada et al., 2019*). This is a typical feature of *for3Δ* cells (*Figure 2—figure supplement 5*; *Okada et al., 2019*). Thus, SAPK-mediated downregulation of For3 levels might be an underlying factor involved in such outcome.

How does the SAPK pathway targets For3 to negatively modulate CAR integrity? This control may not be related with changes in Cdc42 activity, since low LatA concentrations (0.2 µM), which prompt Sty1 activation and inhibit CAR assembly and growth in wild-type cells, did not elicited the dispersal of active Cdc42 from the cell poles observed with higher doses of the drug (*Mutavchiev et al., 2016*). Instead, our results suggest that specific reduction of the formin protein levels is involved, thus representing an additional regulatory layer that limits cell growth and division in response to environmental stress. For3 levels doubled in *sty1Δ* cells, and in those expressing Sty1-T97A in the presence of a specific kinase inhibitor. Conversely, constitutive Sty1 hyperactivation (*wis1DD* or *pyp1Δ* cells) lowered the levels of For3. Increased LatA concentrations also prompted a graded decrease of For3 levels in wild-type cells, but not in *sty1Δ* cells. Moreover, actin cable and CAR perturbation induced by temperature-sensitive alleles of profilin and tropomyosin, without LatA treatment, were accompanied by downregulation of For3 levels that once more was not observed in the absence of Sty1, thus reinforcing the biological relevance of the SAPK pathway as a negative regulator of this formin in response to CAR damage. Sty1-negative regulation of For3 does not occur at the transcriptional level. On the contrary, Sty1 and its downstream target transcription factor Atf1 may elicit *for3+* transcription, since its basal mRNA levels are lower both in *sty1Δ* and *atf1Δ* cells as compared to wild-type cells, and become induced in response LatA. Enhanced expression of the *for3+* gene by Sty1-Atf1 may thus function as a positive feedback mechanism to compensate for the drop in formin levels that follows Sty1 activation.

Ubiquitin pull-down assays performed in wild type and proteasome mutant cells suggest that Sty1 might enhance For3 degradation in response to stress through ubiquitination (*Figure 5—figure*

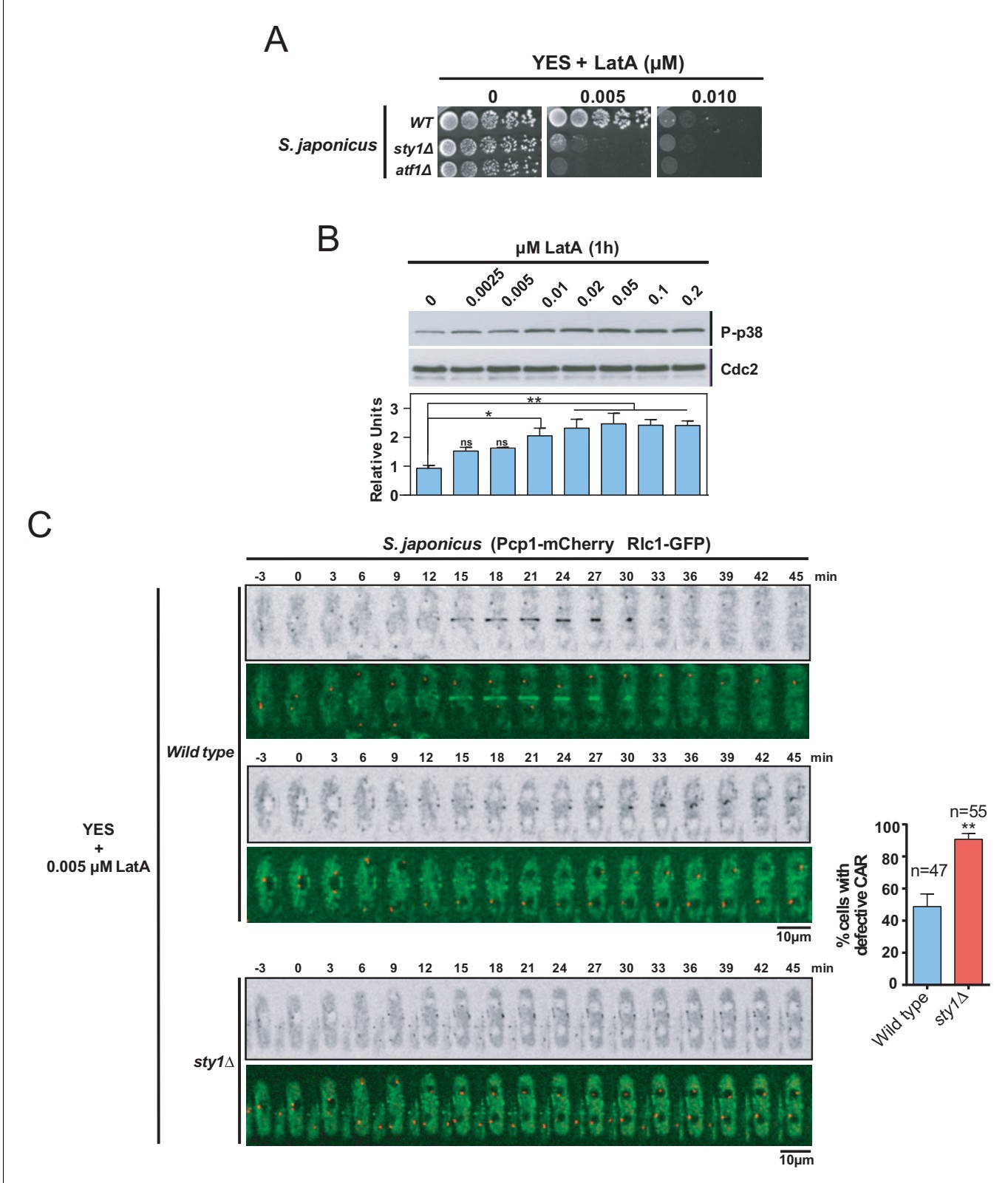

**Figure 8.** The SAPK pathway positively regulates CAR assembly in *S. japonicus* during actin stress induced with LatA. (**A**) Decimal dilutions of *S. japonicus* wild type, *sty1Δ*, and *atf1Δ* strains were spotted on YES and YES solid plates with the indicated concentrations of LatA, incubated at 30°C for 3 days, and photographed. A representative experiment is shown. (**B**) Upper panel: *S. japonicus* wild-type cells were grown in YES medium to mid-log phase, and remained untreated (0), or treated with the indicated concentrations of LatA for 1 hr. Activated Sty1 was detected with anti-phospho-p38

*Figure 8 continued on next page*

*Figure 8 continued*

antibody. Anti-Cdc2 was used as a loading control. Lower panel: Relative units as mean ± SD (biological duplicates) for Sty1 phosphorylation (anti-phospho-p38 blot) were determined with respect to the loading control (anti-Cdc2 blot). **, $p<0.005$; *, $p<0.05$; ns, not significant, as calculated by unpaired Student's *t* test. (C) Representative time-lapse maximum-projection images of Rlc1-GFP dynamics at the equatorial region of *S. japonicus* wild-type and *sty1Δ* cells growing in YES medium to mid-log phase in the presence of 0.005 µM LatA. Mitotic progression was monitored using Pcp1-mCherry-marked SPBs. Time interval is 3 min. Right: the percentage of cells that show damaged/defective CAR was estimated for each strain (n> 47) and is presented as mean ± SD (biological triplicates). ***, $p<0.001$, as calculated by unpaired Student's *t* test.

The online version of this article includes the following source data for figure 8:

**Source data 1.** Values used for graphical representations and statistical analysis in *Figure 8*.

---

*supplement 4*). However, cycloheximide-chase experiments show that For3 has a long half-life (>20 hr; *Figure 5—figure supplement 5*), which is very similar to a value reported in a previous study (t1/2 = 21.3 hr) (*Christiano et al., 2014*). Moreover, Sty1 deletion did not alter For3 half-life as compared to wild-type cells (*Figure 5—figure supplement 5*). Then, how the levels of a 'long-lived' protein like For3 become quickly reduced during stress? The switch of For3 from an autoinhibited (closed) to an active (open) conformation depends on the relieving of the intramolecular interaction between its functional DAD and DID domains, and is essential for its stabilization and localization at the CAR and the cell poles (*Martin et al., 2007*). We hypothesize that during vegetative growth, most For3 is active (open) and highly stable, while in response to stress SAPK activation might trigger its rapid ubiquitination and degradation. The mechanism/s responsible for this modification might include the switch of the formin to a closed and unstable conformation, and/or the activation of specific ubiquitin ligase/s either at a transcriptional or post-transcriptional level. For3 is heavily phosphorylated in vivo (*Kettenbach et al., 2015*; *Swaffer et al., 2018*), but its phosphorylation status did not change along the cell cycle. The kinase/s responsible for such modification/s is/are currently unknown, but the possibility that Sty1 phosphorylates For3 seems unlikely, since it was not phosphorylated by Sty1 in vitro (*Figure 5—figure supplement 6*), and its phosphorylation pattern remained unaffected in vivo independently of the kinase activation status (*Figure 5*). Further work will be necessary to depict the precise mechanism/s by which the SAPK pathway negatively modulates For3 levels.

In mammalian cells, formin (mDIA)-induced nuclear actin polymerization promotes activation of the serum response factor (SRF), a key regulator of the actin cytoskeleton dynamics (*Baarlink et al., 2013*). SRF elicits transcriptional activation of multiple genes involved in actin cytoskeleton homeostasis by the myocardin-related transcription factors (MRTFs), whose nuclear accumulation is positively regulated upon release from G-actin binding during filamentous actin polymerization (*Miano et al., 2007*). Similar to the SAPK-For3 branch described here, a putative stress-mediated inhibition of mDIA availability and/or function could play a role to inhibit SRF transcriptional activity by favoring binding of MRTFs to G-actin. It would also be interesting to analyze if the increase in the levels of monomeric G-actin resulting from For3 inhibition by SAPK during environmental stress downregulates the transcriptional activity of SRF orthologs (Mbx1, Map1) in fission yeast.

Our preliminary data suggest that the activation of the SAPK pathway fosters CAR assembly in *S. japonicus* during cytoskeletal damage with LatA. The reason for the opposite effect of SAPK signaling on CAR assembly in *S. japonicus* and *S. pombe* might be related to the marked differences in the timely coordination of mitosis and cytokinesis in both fission yeast species. In *S. pombe,* cortical actin nucleation closely follows myosin recruitment to the medial nodes at the onset of mitosis, whereas in *S. japonicus* actin filaments appear at the cortex late during exit from mitosis (*Gu and Oliferenko, 2015*). SAPK-mediated downregulation of CAR integrity through For3 may represent in *S. pombe* an alternative mechanism to prompt cytokinesis delay until precise chromosome segregation has been secured. Nonetheless, the opposite effect of SAPK pathway activation on CAR assembly and integrity within the fission yeast clade likely illustrates how a strongly conserved signaling cascade may adopt different strategies to cope with the distinctive cell cycle and division characteristics in evolutionarily related species.

# Materials and methods

## Key resources table

| Reagent type (species) or resource | Designation | Source or reference | Identifiers | Additional information |
|---|---|---|---|---|
| Antibody | Anti-Phospho-p38 (rabbit polyclonal) | Cell Signaling | Cat# 9211, RRID:AB_331641 | WB (1:1000) |
| Antibody | Anti-HA (mouse monoclonal) | Roche | Cat# 11 583 816 001, RRID:AB_514505 | WB (1:1000) |
| Antibody | Anti-GFP (mouse monoclonal) | Roche | Cat# 11 814 460 001, RRID:AB_390913 | WB (1:1000) |
| Antibody | HRP-conjugated anti-HA antibody (rat monoclonal) | Roche | Cat# 12 013 819 001, RRID:AB_390917 | WB (1:3000) |
| Antibody | Anti-Cdk1/Cdc2 (PSTAIR) (rabbit polyclonal) | Millipore | Cat#: 06–923; RRID:AB_310302 | WB (1:1000) |
| Antibody | Anti-beta actin (mouse monoclonal) | Abcam | Cat#: ab8224; RRID:AB_449644 | WB (1:4000) |
| Antibody | Anti-Atf1 (ATF1 2A9/8) (mouse monoclonal) | Abcam | Cat#: ab18123; RRID:AB_444264 | WB (1:2000) |
| Antibody | Anti-Mouse IgG- peroxidase (goat polyclonal) | Sigma Aldrich | Cat#: A5278; RRID:AB_258232 | WB (1:2000) |
| Antibody | Anti-Rabbit IgG- peroxidase (goat polyclonal) | Sigma Aldrich | Cat#: A6667; RRID:AB_258307 | WB (1:2000) |
| Antibody | Anti-Thiophosphate ester (rabbit monoclonal) | Abcam | Cat#: ab239919 | WB (1:5000) |
| Antibody | Anti-GST HRP-conjugated (goat polyclonal) | GE Healthcare | Cat#: RPN1236; RRID:AB_771429 | WB (1:5000) |
| Antibody | Anti-ubiquinin HRP-conjugated (mouse monoclonal) | Santa Cruz Biotechnology | Cat#: sc-8017; RRID:AB_628423 | WB (1:2000) |
| Commercial assay, kit | RNeasy mini kit | Quiagen | Cat#: 74104 | |
| Commercial assay, kit | iScript reverse transcription supermix | Bio-Rad | Cat#: 1708841 | |
| Commercial assay, kit | iTaq Universal SYBR Green Supermix | Bio-Rad | Cat#: 1725120 | |
| Commercial assay, kit | ECL Western Blotting Reagents | GE-Healthcare | Cat#: RPN2106 | |
| Chemical compound, drug | β-estradiol | Sigma Aldrich | Cat#: E2758 | 10–500 µM |
| Chemical compound, drug | PhosTag acrylamide | Wako Chemical | Cat#: 300–93523 | 15 µM |
| Chemical compound, drug | BrB-PP1 AKT inhibitor | Abcam | Cat#: ab143756 | 20 µM |
| Chemical compound, drug | PP1 Analog III, 3-MB-PP1 | Sigma Aldrich | Cat#: 529582 | 1 µM |
| Chemical compound, drug | Latrunculin A | Wako Chemical | Cat#: 129–04361 | 0.05–3 µM |
| Chemical compound, drug | Alexa fluor 488-conjugated phalloidin | Thermo Fischer Scientific | Cat#: A12379 | 200 units/ml (~6.6 µM) |
| Chemical compound, drug | Soybean lectin | Sigma Aldrich | Cat#: L2650 | 1 mg/ml |
| Software, algorithm | ImageJ | ImageJ | https://imagej.net/Fiji/Downloads | Quantification of western blots and microscopic analysis |

*Continued on next page*

*Continued*

| Reagent type (species) or resource | Designation | Source or reference | Identifiers | Additional information |
|---|---|---|---|---|
| Software, algorithm | Graphpad Prism 6.0 | Graphpad | https://www.graphpad.com/scientific-software/prism// | Statistical analysis and graphs representation |
| Other | µ-Slide eight well | Ibidi | Cat#: 80826 | |

## Fission yeast strains, growth conditions and reagents

The *S. pombe* and *S. japonicus* strains used in this work are listed in *Supplementary file 1*. They were routinely grown with shaking at 28°C or 30°C in 0.6% yeast extract with 2% glucose, and supplemented with adenine, leucine, histidine, or uracil (100 mg/L, Sigma-Aldrich) (YES medium) (*Moreno et al., 1991*). Solid medium was made by adding 2% agar (Difco Bacto Agar). Minimal (EMM2) medium (*Moreno et al., 1991*) with or without 5 µg/ml thiamine was employed in ubiquitin pull-down experiments. Strains expressing different genomic fusions in single and multiple genetic backgrounds were constructed either by transformation, or after tetrad or random spore dissection and analysis of appropriate crosses in sporulation medium (SPA) (*Petersen and Russell, 2016*). Latrunculin A (Wako Chemical) was added from a filter-sterilized stock (5 mM in DMSO) at the indicated concentrations to YES log-phase liquid cultures (*S. pombe*: $OD_{600} = 0.5$; ~$10^6$ cells/ml; *S. japonicus*: $OD_{600} = 0.5$; ~$2.25 \ 10^5$ cells/ml), or to solid medium after autoclaving. To investigate protein degradation rate, log-phase liquid cultures were incubated in the presence of 100 µg/ml of the translational inhibitor cycloheximide (Sigma Aldrich). In experiments with cells expressing an analogue-sensitive cdc2 kinase version (cdc2-asM17), log-phase liquid cultures were treated with 1 µM 3-MB-PP1 (Sigma Aldrich, 529582) dissolved in DMSO. In experiments with strains expressing an analogue-sensitive Sty1 kinase version (Sty1-T97A), log-phase liquid cultures were treated with 10 µM 3BrPP1 (Abcam, ab143756) prepared from a 25 mM stock dissolved in methanol.

## Expression of For3 under the control of the β-estradiol-regulated promoter

To obtain a plasmid expressing a For3-HA C-terminal fusion under the control of the β-estradiol-regulated promoter, the $Z_3$EV promoter sequence first was amplified from plasmid pFS462 (*Ohira et al., 2017*) using oligonuclotides XhoI-Z3EVPR-F and SmaI-Z3EVPR-R (*Supplementary file 2*). The PCR product was digested with *XhoI* and *SmaI* and cloned into integrative plasmid pJK210 (*Petersen and Russell, 2016*). Next, *for3+* ORF was amplified by PCR using genomic DNA from wild-type cells as template and the oligonucleotides For3-PJK210ZEV-F(SmaI) and For3-HA-PJK210ZEV-R(SacII), which incorporates a DNA sequence encoding a single HA epitope in the C-terminal end of the ORF (*Supplementary file 2*). The purified PCR product was digested with *SmaI* and *SacII* and cloned into pJK210-$Z_3$EVpr to obtain plasmid pJK210-$Z_3$EVpr:For3-HA, which was confirmed by DNA sequencing. Plasmid pJK210-$Z_3$EVpr:For3-HA was then digested within the *ura4+* ORF with *StuI* and transformed into strain E1723 (*for3Δ ura4.294*) (*Supplementary file 1*) to yield strain E1751 (*for3Δ $Z_3$EVpr*:For3-HA:ura4+ *ura4.294*). Finally, strain E1751 was crossed with the strain E1687, which constitutively expresses the $Z_3$EV transcription factor under the control of the strong *adh1* promoter (*Ohira et al., 2017*), yielding the inducible strain E1764 (*for3Δ adh1*: $Z_3$EV *$Z_3$EVpr*: For3-HA). In experiments performed with this strain, cultures growing in either liquid or solid YES medium were treated with varying amounts of β-estradiol (Sigma Aldrich, E2758), prepared from a 5 mM stock dissolved in methanol.

## cDNA synthesis and quantitative real-time polymerase chain reaction (qPCR)

*S. pombe* wild-type and mutant strains were grown in YES medium to a final $OD_{600} = 0.5$; (~$10^6$ cells/ml). Total RNAs were purified using the RNeasy mini kit (Qiagen), treated with DNase (Invitrogen), and quantitated using Nanodrop 100 spectrophotometer (ThermoScientific). Total RNAs (1 µg) were reverse transcribed into cDNA with the iScript reverse transcription supermix (BioRad). Quantitative real time polymerase chain reactions (qPCR) were performed using the iTaq Universal SYBR Green Supermix and a CFX96 Real-Time PCR system (BioRad Laboratories, CA). Relative gene

expression was quantified based on $2^{-\Delta\Delta CT}$ method and normalized using $leu1^+$ mRNA expression in each sample. The list of gene-specific primers for qPCR is indicated in *Supplementary file 2*.

## Detection of total and activated Sty1 levels

We followed the method described in *Mutavchiev et al., 2016* with slight modifications. Fission yeast cultures were grown in YES to a final $OD_{600}$ = 0.5, and were supplemented with the desired final concentrations of Lat A or 1% DMSO (solvent control). Samples of 5 ml were collected at the indicated times and immediately centrifuged for 20 s at 3200 rpm/4°C. The cell pellets were resuspended in 1 ml ice-cold 10 mM NaPO4 0.5 mM EDTA pH 7.5 buffer, transferred to 1.5 ml tubes, centrifuged at 13,000 rpm/4°C, and stored at −80°C until further processing. Cell lysis was achieved in a FastPrep instrument after mixing the cell pellets with pre-chilled 0.5 mm glass beads to −20°C with ice-cold lysis buffer (20 mM Tris-HCl, pH 8.0, 2 mM EDTA, 100 mM NaCl, and 0.5% NP-40 and containing a protease inhibitor cocktail [Sigma Aldrich]). The cell lysates were finally clarified by centrifugation at 13,000 rpm at 4°C for 5 min. Dual phosphorylation of Sty1 was detected employing a rabbit polyclonal anti-phospho-p38 antibody (RRID:AB_331641; Cell Signaling). Total Sty1 was detected in *S. pombe* extracts with mouse monoclonal anti-HA antibody (RRID:AB_514505; 12CA5, Roche Molecular Biochemicals), whereas rabbit polyclonal anti-Cdk1/Cdc2 (PSTAIR) (RRID:AB_310302; Millipore) was used as loading control in *S. japonicus* extracts. Immunoreactive bands were revealed with anti-mouse (RRID:AB_258232) or anti-rabbit (RRID:AB_258307) HRP-conjugated secondary antibodies (Sigma Aldrich) and the ECL system (GE-Healthcare).

## Detection of For3 levels

Total protein extracts from exponentially growing *S. pombe* cultures ($OD_{600}$ = 0.6) expressing a genomic For3-3GFP fusion were obtained with lysis buffer (20 mM Tris-HCl, pH 8.0, 2 mM EDTA, 100 mM NaCl, and 0.5% NP-40 and containing a protease inhibitor cocktail [Sigma-Aldrich]) as described above. Proteins were resolved in 6% SDS-PAGE gels and transferred to Hybond-ECL membranes. To detect For3 in synchronized cultures Cdc25-22 cells expressing For3-3GFP were grown in YES to mid log phase, shifted to 36.5°C for 3.5 hr, and then released to the permissive temperature (25°C). Samples were collected every 30 min and the corresponding protein extracts were obtained. Cell samples were also collected in parallel at each time-point, fixed with cold 70% ethanol, and monitored for cell cycle progression and septation after DAPI and calcofluor staining, respectively. Cdc10-129 (G1-phase arrest), and Nda3-km311 (M-phase arrest) mutants expressing For3-3GFP genomic fusion were incubated at either 36.5°C for 3.5 hr (*cdc10-129* background) or 18°C for 7 hr (*nda3-km311* background). The For3-3GFP fusion was detected an all experiments the employing a mouse monoclonal anti-GFP antibody (RRID:AB_390913; Roche). Detection of the For3-HA fusion under the control of the β-estradiol-regulated promoter (Z3EVpr) was achieved by incubation with a rat monoclonal HRP-conjugated anti-HA antibody (RRID:AB_390917; clone 3F10, Roche). Rabbit polyclonal anti-Cdk1/Cdc2 (PSTAIR) (RRID:AB_310302; Millipore) was used for loading control. Immunoreactive bands were revealed with anti-rabbit peroxidase-conjugated secondary antibody (RRID:AB_258307; Sigma Aldrich), and the ECL system (GE-Healthcare).

## Lambda phosphatase treatment

In the For3 dephosphorylation assay 10 μg of protein were treated with 40 U of λ-protein phosphatase (New England Biolabs) in the presence/absence of specific phosphatase inhibitor (5 mM sodium orthovanadate) for 50 min at 30°C. Protein electrophoresis was performed on 6% SDS-PAGE gels and the For3-3GFP fusion was detected as indicated.

## Ubiquitin pull-down experiments

Expression of nmt1 promoter-driven 6His-ubiquitin (pREP1-6His-Ubi) was performed by growing cells in liquid EMM2 -leu or -ura media without thiamine for 24 hr. Forty-five minutes prior to harvest, cells were treated with 5 mM N-ethylmaleimide (NEM; Sigma Aldrich) added directly to the growth medium. Then, exponentially growing cultures corresponding to an $OD_{600}$ = 0.8–1 were harvested and the cells were washed once with ice-cold PBS 1x. Cell lysis was achieved in a FastPrep instrument after mixing the cell pellets with pre-chilled 0.5 mm glass beads with ice-cold buffer G (6 M guanidine hydrochloride, 0.1 M sodium phosphate, 50 mM Tris-HCl pH 8). The cell lysates were

finally clarified by centrifugation at 13,000 rpm at 4°C for 10 min. His-Ubiquitin conjugates were puri-fied by incubating with $Ni^{++}$-nitrilotriacetic acid-agarose beads (Qiagen) for 2 hr at 4°C. The bound material was washed three times with a buffer containing 20 mM Tris-HCl, pH 8.0, 2 mM EDTA, 100 mM NaCl, 0.5% Nonidet P-40, and 15 mM imidazole, plus a protease inhibitor cocktail (Sigma Aldrich). The purified proteins were eluted with sample buffer, incubated at 100°C for 8 min, resolved through 6% SDS-PAGE gels, transferred to Hybond-ECL membranes, and analyzed by immunoblotting with a mouse monoclonal anti-GFP (RRID:AB_390913; Roche) and anti-ubiquitin HRP-conjugated (RRID:AB_628423; Santa Cruz Biotechnology) antibodies. The immunoreactive bands were revealed with anti-mouse peroxidase-conjugated secondary antibody (RRID:AB_258232; Sigma-Aldrich), and the ECL system (GE-Healthcare).

## Detection of Cdc12, Atf1, and actin levels

Cells from yeast cultures were fixed and total protein extracts were prepared by precipitation with trichloroacetic acid (TCA) as previously described (*Grallert and Hagan, 2017*). Proteins were resolved in 6% (Cdc12) or 10% (Atf1; actin) SDS-PAGE gels and transferred to Hybond-ECL membranes. Where indicated, samples were resolved by SDS-PAGE in the presence of 15 µM PhosTag acrylamide (Wako Chemical) following the indications of the manufacturer. Cdc12-3HA fusions were detected employing a rat monoclonal HRP-conjugated anti-HA antibody (RRID:AB_390917; clone 3F10, Roche). Actin and Atf1 levels were detected, respectively, with mouse monoclonal anti-beta actin (RRID:AB_449644; ab8224) and anti-Atf1 (RRID:AB_444264; ab18123) antibodies from Abcam. Rabbit polyclonal anti-Cdk1/Cdc2 (PSTAIR) (RRID:AB_310302; Millipore) was used for loading con-trol. Immunoreactive bands were revealed with anti-mouse (RRID:AB_258232) or anti-rabbit (RRID: AB_258307) HRP-conjugated secondary antibodies (Sigma Aldrich) and the ECL system (GE-Healthcare).

## In vitro kinase assay

GST-Wis1DD (constitutively active MAPKK), GST-Sty1 (T97A) (analogue-sensitive MAPK), and GST-Atf1 fusions were expressed and purified from *Escherichia coli* with glutathione-Sepharose 4B beads (GE Healthcare, USA) as described (*Mutavchiev et al., 2016*). Bacterially expressed fusions GST-For3$_1$ (aminoacids 1–390), GST-For3$_2$ (aminoacids 391–920) and GST-For3$_3$ (aminoacids 921–1461) (*Figure 5—figure supplement 2*) were obtained by PCR employing genomic DNA from wild type cells as a template, and the respective oligonucleotide pairs GST-For31-F(SmaI) plus GST-For31-R (NcoI), GST-For32-F(SmaI) plus GST-For32-R(NcoI), and GST-For33-F(SmaI) plus GST-For33-R(NcoI) (*Supplementary file 2*). The corresponding PCR products were then digested with *Sma*I and *Nco*I and cloned into plasmid pGEX-KG. Once purified and after extensive washing, GST-Wis1DD, GST-Sty1 (T97A), plus GST-For3$_1$, GST-For3$_2$, GST-For3$_3$, or GST-Atf1 fused substrates were incubated in 20 mM Tris (pH 8), 10 mM $MgCl_2$, and 20 µM ATPγS at 30°C for 45 min in the presence/absence of 20 µM kinase-specific inhibitor BrB-PP1 (Abcam). The kinase reaction was stopped by adding 20 mM EDTA, and the reaction mixture was alkylated after incubation at room temperature with 2.5 mM *p*-nitrobenzyl mesylate for 1 hr. Atf1 and/or For3 phosphorylation was detected with rabbit monoclo-nal anti-Thiophosphate ester antibody (ab239919; Abcam). GST fusions were detected with goat anti-GST HRP-conjugated antibody (RRID:AB_771429; GE Healthcare).

## Quantification and reproducibility of western blot experiments

Densitometric quantification of western blot signals as of 16-bit. jpg digital images of blots was per-formed using ImageJ (*Schneider et al., 2012*). The desired bands plus background were drawn as rectangles and a profile plot was obtained for each band (peaks). To minimize the background noise in the bands, each peak floating above the baseline of the corresponding peak was manually closed off using the straight-line tool. Finally, measurement of the closed peaks was performed with the wand tool. Relative Units for Sty1 activation in *Figures 1C* and *8B* were estimated by determining the signal ratio of the anti-phospho-p38 blots with respect to the anti-HA blot (total Sty1 in *S. pombe* extracts) or anti-cdc2 (*S. japonicus* extracts) at each time point. Relative Units for total Act1 and For3 levels in *Figure 5B, C, F, G* and *Figure 7B*, *Figure 2—figure supplement 2*, *Figure 5—fig-ure supplement 2* and *Figure 5—figure supplement 6* were estimated by determining the signal ratio of the correspondent anti-Act1, or anti-GFP (total For3) blots with respect to the anti-cdc2 blot

(internal control) at each time point. Depending on the experiment, quantification data shown may correspond to representative experiments, or those performed as biological duplicates or triplicates. Mean relative units ± SD and/or representative results are shown. p-Values were analyzed by unpaired Student's *t* test. Graphs and statistical analyses were performed with Prism 6.0 (GraphPad Software).

## Plate assays of stress sensitivity for growth

*S. pombe* and *S. japonicus* wild-type and mutant strains were grown in YES liquid medium to $OD_{600}$ of 0.5 (*S. pombe*) or 0.3 (*S. japonicus*), and appropriate decimal dilutions were spotted per triplicate on YES solid plates or in the same medium supplemented with varying concentrations of LatA. Plates were incubated at either 28 or 30°C for 3 days and then photographed. All the assays were repeated at least three times with similar results. Representative experiments are shown in the corresponding Figures.

## Microscopy analysis

For calcofluor staining, a solution of Calcofluor White (50 mg/ml final concentration) was directly added to early logarithmic phase cells. To perform actin staining with Alexa-Fluor phalloidin, mid-log cultures in YES (5 ml) were fixed for 40 min at 28°C with 1/10 vol of PEM (0.1 M Na PIPES pH 6.8, 1 mM EGTA, 1 mM $MgCl_2$), and 1/5 vol of 16% EM-grade formaldehyde. Cells were then washed three times with 0.5 ml PEM, permeabilized for 30 s with PEM/1% Triton X-100, washed three additional times with PEM, and the cell pellets were resuspended in the remaining liquid. For staining, 8 µl of Alexa fluor 488-conjugated phalloidin (Thermo Fisher Scientific) dissolved in PEM (~6.6 µM) were added to 1 µl of the resuspended cell pellets, and incubated in the dark for 1 hr at room temperature in a rocking platform.

Fluorescence images in *Figures 1G*, *2F*, *3C* and *Figure 7B* were obtained with a Leica DM4000B microscope equipped with a Leica DC400F camera, and processed using IM500 Image Manager software. Fluorescence images in *Figure 2—figure supplement 4* and *Figure 5D* were obtained with a Personal DeltaVision System (GE Healthcare), and corrected by 3-D deconvolution (conservative ratio, 10 iterations, and medium noise filtering) using the SoftWoRx imaging software (6.0, GE Healthcare). Line scans of For3-GFP or Pmk1-GFP cells (n> 30) stained with Calcofluor (*Figure 5D*) were made by drawing a line along the equatorial region in cells with early septum (less that 0.7 µm), and measuring the intensities of pixels of the green fluorescence through the plot profile function of the ImageJ software (*Mutavchiev et al., 2016*). Images in *Figure 2A* were acquired using a spinning disk confocal microscope (Olympus IX81 with Roper technology) with a 100X/1.40 Plan Apo lens and controlled by Metamorph 7.7 software (Molecular Devices, Sunnyvale, CA).

For *time-lapse* imaging experiments, 0.3 ml of early log-phase cell cultures (*S. pombe*: $OD_{600} = 0.5$; *S. japonicus*: $OD_{600} = 0.3$) were placed in a well from a µ-Slide eight well (Ibidi) previously coated with 10 µl of 1 mg/ml soybean lectin (Sigma-Aldrich) (*Cortés et al., 2012*). Cells were left for 1 min to attach to the well bottom and culture media was carefully removed. Then, cells were washed three times with the same media and finally 0.3 ml of fresh media with LatA or without the drug (DMSO; solvent untreated control) were added. Experiments were performed at 28°C and single middle planes from a set of five stacks (0.61 µm each) were taken at the time points indicated in each figure. Time-lapse images in *Figures 2D, E*, *3B*, *4B, C* and *8C*, *Figure 2—figure supplement 5*, and *Figure 3—figure supplement 4* were acquired using a Leica SP8 confocal microscope with a 63X/1.40 Plan Apo objective and controlled by the LAS X software. In *Figure 2—figure supplement 5* the time for node condensation and ring maturation includes the time from SPB separation until the start of CR constriction. The time for ring constriction and disassembly includes the time from the first frame of ring constriction until the last frame where it becomes completely constricted and disassembled. The extent of ring damage during LatA treatment was determined in time-lapse experiments by quantifying the percentage of cells in which CAR condensation from cytokinetic nodes is strongly compromised or blocked. In asynchronous culture experiments, CAR damage was expressed as the percentage of total cells with rings showing morphologically altered, and/or aberrantly condensed rings.

Graphs and statistical analysis were done using Graphpad Prism 5.0 (GraphPad Software). Unless otherwise stated, graphs represent the mean, and error bars represent SD. *n* is the total number of

cells scored from at least three independent experiments. Statistical comparison between two groups was performed by unpaired Student's *t* test.

## Acknowledgements

We thank Mohan Balasubramanian, Kathy Gould, Tomohisa Kato Jr., Jonathan Millar, Hironori Niki, Snezhana Oliferenko, Miguel Angel Rodriguez-Gabriel, Kazuhiro Shiozaki, and Takashi Toda for fission yeast strains and plasmids, and to Rafael R Daga for stimulating discussions. This work was supported by the Ministerio de Ciencia, Innovación y Universidades, Spain [Grant references BFU2017-82423-P and PGC2018-098924-B-100], Fundación Séneca de la Región de Murcia [Grant reference 20856/PI/18], and Junta de Castilla y León 'Escalera de la Excelencia', Spain [Grant reference CLU-2017–03]. European Regional Development Fund (ERDF) co-funding from the European Union. E G-G and F P-R are Formación de Profesorado Universitario (FPU) pre-doctoral fellows from the University of Murcia and the Ministerio de Ciencia, Innovación y Universidades, Spain, respectively.

## Additional information

### Funding

| Funder | Grant reference number | Author |
| --- | --- | --- |
| Ministerio de Economía y Competitividad | BFU2017-82423-P | Jose Cansado |
| Ministerio de Economía y Competitividad | PGC2018-098924-B-100 | Pilar Pérez |
| Junta de Castilla y Leon | CLU-2017-03 | Pilar Pérez |
| Fundacion Seneca | 20856/PI/18 | Jose Cansado |

The funders had no role in study design, data collection and interpretation, or the decision to submit the work for publication.

### Author contributions

Elisa Gómez-Gil, Rebeca Martín-García, Data curation, Formal analysis, Investigation, Methodology; Jero Vicente-Soler, Formal analysis, Investigation, Methodology, Project administration; Alejandro Franco, Beatriz Vázquez-Marín, Francisco Prieto-Ruiz, Formal analysis, Investigation, Methodology; Teresa Soto, Formal analysis, Supervision, Investigation, Methodology; Pilar Pérez, Conceptualization, Resources, Supervision, Funding acquisition, Validation, Visualization, Project administration, Writing - review and editing; Marisa Madrid, Conceptualization, Data curation, Formal analysis, Supervision, Validation, Visualization, Writing - review and editing; Jose Cansado, Conceptualization, Resources, Formal analysis, Supervision, Funding acquisition, Validation, Investigation, Writing - original draft, Writing - review and editing

### Author ORCIDs

Teresa Soto (iD) http://orcid.org/0000-0003-2965-318X
Jose Cansado (iD) https://orcid.org/0000-0002-2342-8152

### Decision letter and Author response

Decision letter https://doi.org/10.7554/eLife.57951.sa1
Author response https://doi.org/10.7554/eLife.57951.sa2

## Additional files

### Supplementary files

- Supplementary file 1. *S. pombe* and *S. japonicus* strains used in this study.
- Supplementary file 2. Oligonucleotides employed in this study.

• Transparent reporting form

### Data availability

All data generated or analysed during this study are included within the manuscript and supporting files.

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
