## [Decision Letter]

**Acceptance summary:**

This is an interesting manuscript from Cansado and colleagues on the negative regulation of actomyosin ring integrity upon actin stress. The message is that upon actin stress, formin-for3 levels drop in a ubiquitination and stress activated protein kinase dependent pathway, which leads to a cytokinesis delay and growth advantage following actin stress removal. The work has potential to open up new lines of regulation of formin and actin cytoskeleton by stress-activated signalling.

**Decision letter after peer review:**

Thank you for submitting your article "Stress-activated MAPK signalling controls fission yeast actomyosin ring integrity by modulating formin For3 levels" for consideration by *eLife*. Your article has been reviewed by three peer reviewers, one of whom is a member of our Board of Reviewing Editors, and the evaluation has been overseen by Anna Akhmanova as the Senior Editor. The following individual involved in review of your submission has agreed to reveal their identity: Thomas D Pollard (Reviewer #2).

The reviewers have discussed the reviews with one another and the Reviewing Editor has drafted this decision to help you prepare a revised submission.

The reviewers and the editor found the work and the new insights into the regulation of formin for3 stability and cytokinesis by the Stress Activated MAPK pathway upon LatA stress very interesting. A large number of issues have been raised, which are being relayed to you verbatim, so that you may be able to investigate them for subsequent work (due to the COVID19 policy).

Essential revisions for this paper:

We see three experiments as essential before publication.

1) Mechanisms: The big question is why cells treated with LatA reduce the level of For3 protein in 2 hours in spite of a parallel increase in For3 mRNA levels? The mechanisms are not addressed. The authors should consider a pulse-chase experiment to measure the turnover of For3 protein. The Discussion suggests that "Sty1-Atf1 branch or Sty1 alone might favor For3 degradation via ubiquitination" but the gels do not show higher molecular weight bands.

2) The over-reliance on using low doses of LatA as the sole source of actin-cable stress is of significant concern. Perhaps the authors can investigate the effects of other mutants that affect actin cable integrity (*cdc8* tropomyosin, *cdc3* profilin etc) on for3 levels as well as estimate for3 levels in *cdc8sty1Δ* and *cdc3sty1Δ* mutants. This will be a nice addition to the paper.

3) The resistance of *sty1* mutants to low doses of LatA raises a simple possibility that LatA does not enter the s*ty1Δ* cell as readily as in WT cells. This could, for instance, be due to potential differences in the cell wall. If true, it would be a major caveat. One suggested test may be to assay dose dependent responses of LatA on actin patches, for instance.

Revisions expected in follow-up work:

1) A number of quantitation experiments have been done with asynchronously growing cells (Figures 2B, 3E for example, and there are more). Some are backed up by time-lapse imaging with SPB markers, and others not. This needs to be looked at carefully throughout the work to make sure the phenotypes are neither a overestimate or underestimate.

2) It is not clear how the authors explain the fact that the drop in *for3* levels happens very slowly, whereas the phenotypic defects happen in minute timescales. It will be informative to hold the cell cycle using a *cdc25-22* or *nda3-KM311* and investigate the reduction and increase of *for3* levels upon LatA treatment. It is important to know whether the cytokinesis phenotype is a comorbidity or the main effect of LatA induced reduction in *for3* levels.

3) This work would gain in significance if it explored more about the mechanism of how Sty1 pathway affects For3 proteins levels. Could it be regulating translation or stability of the protein? For instance, the authors could test whether Sty1 affects For3 stability using a cycloheximide experiment and/or measure For3 translation.

4) For3 is best known for its role in actin cable assembly. Given that Sty1 affects For3 levels even in interphase cells, the authors should investigate how stress and Sty1 affect actin cables as well. The images of actin shown in Figure 2 suggest that actin cables in *sty1Δ* cells look brighter; effects on actin cables could be quantitated. Do other stresses and the Sty1 pathway impact actin cable assembly, and could this also be suppressed by formin activation or overexpression. Is there a reason why stress pathway should inhibit actin cables?

---

## [Author Response]

Essential revisions for this paper:We see three experiments as essential before publication.1) Mechanisms: The big question is why cells treated with LatA reduce the level of For3 protein in 2 hours in spite of a parallel increase in For3 mRNA levels? The mechanisms are not addressed. The authors should consider a pulse-chase experiment to measure the turnover of For3 protein. The Discussion suggests that "Sty1-Atf1 branch or Sty1 alone might favor For3 degradation via ubiquitination" but the gels do not show higher molecular weight bands.

Response also to comment #3 of Revisions expected in follow-up work.

As suggested by the editors and the reviewers, we determined the half-life of a genomic For3-3GFP fusion by blocking protein synthesis with cycloheximide and measuring the decline in the protein levels of the formin at different time points. As can be seen in the Figure 5—figure supplement 5 in the revised manuscript, cyclohexymide treatment doubled For3 levels in wild type cells during the first 12 h of incubation, which started to decrease gradually at longer incubation times. The estimated half-life of For3 was long (>20 hours), which fairly agrees with that obtained from a global analysis of proteome turnover in fission yeast that categorized For3 as a stable “class III” protein (t_1/2_>8h; For3 t_1/2_ =21.3h) (Christiano et al., 2014). However, Sty1 deletion did not have an evident effect on For3 half-life, which was quite similar to that of wild type cells:

Next, we analyzed the possibility that For3 becomes ubiquitinated in vivo. To this end we performed ubiquitin pull-down assays with the pREP-Ubi-His plasmid in wild type cells and the temperature-sensitive mutant of the proteasome subunit Mts3/Rpn12 (*mts3-1* cells), expressing a genomic For3-3GFP fusion. As it can be seen in the panel A in Figure 5—figure supplement 4 in the revised manuscript, ubiquitinated For3 species were specifically recovered from wild type cells during unperturbed growth (28ºC).

Importantly, as compared to wild type cells, the For3 ubiquitination ratio increased significantly in *mts3-1* cells incubated at the restrictive temperature of 36ºC (panel B). New text and the corresponding supplementary figures describing these findings have been included in the Results section of the revised manuscript to read as follows:

“mDia2 formin, a key nucleator of actin filaments during cell migration in mammalian cells, is degraded at the end of mitosis through ubiquitination, and this modification is essential for the completion of cell division [DeWard and Alberts, 2009]. […] Considering that the decrease in total For3 levels at high temperature is Sty1-dependent (Figure 5—figure supplement 3), these findings suggest that a SAPK-mediated mechanism enhances For3 in vivo ubiquitination and degradation in response to stress.”

Previous work (Martin et al., 2007), showed that the switch of For3 from an autoinhibited (closed) to an active (open) conformation, which depends on the relieving of the intramolecular interaction between its functional DAD and DID domains, is essential for For3 stabilization and localization at the CAR and cell poles. Based on this precedents and our novel findings, we propose that during vegetative growth For3 is mainly active (open) and stable (long half-life), while SAPK activation may trigger its rapid ubiquitination and degradation (short half-life). Several mechanisms, including hampering of For3 activation or the activation of the ubiquitination machinery, might be involved in the Sty1-induced formin destabilization. This reasoning has been included in the Discussion section of the revised manuscript:

“Ubiquitin pull-down assays performed in wild type and proteasome mutant cells suggest that Sty1 might enhance For3 degradation in response to stress through ubiquitination (Figure 5—figure supplement 4). […] The mechanism/s responsible for this modification might include the switch of the formin to a closed and unstable conformation, and/or the activation of specific ubiquitin ligase/s either at a transcriptional or post-transcriptional level.”

Although the specific mechanism responsible for the Sty1-dependent decrease in For3 levels is not depicted, we honestly believe that the establishment of a putative link between the SAPK pathway and For3 ubiquitination status improves the quality of the results in the current work, and opens new lines of research for a full understanding of the nature of this complex functional interplay.

2) The over-reliance on using low doses of LatA as the sole source of actin-cable stress is of significant concern. Perhaps the authors can investigate the effects of other mutants that affect actin cable integrity (cdc8 tropomyosin, cdc3 profilin etc) on for3 levels as well as estimate for3 levels in cdc8 sty1Δ and cdc3 sty1Δ mutants. This will be a nice addition to the paper.

Following the recommendation of the editor and reviewers, we have investigated the biological effects on For3 levels of hypomorphic mutations in profilin (*cdc3-124*) and tropomyosin (*cdc8-110*), which play a positive role in maintenance of actin filaments for the CAR during cytokinesis.

As compared to wild type cells, For3 levels were reduced in *cdc3-124* and *cdc8-110* mutants incubated at the restrictive temperature (Figure 6B). Moreover, Sty1 deletion partially suppressed the thermosensitivity of *cdc3-124* cells while enhancing For3 levels in both mutants. These findings reveal the general significance of the SAPK-For3 branch during control of actin cables and CAR integrity in response to stress (Figure 6A and B). In addition, text and references have been included in the Results section to read:

“The actin-binding proteins profilin (Cdc3) and tropomyosin (Cdc8) localize to the contractile ring and play essential roles in the formation, stabilization, and maintenance of actin filaments for the CAR during cytokinesis [Balasubramanian, Helfman and Hemmingsen, 1992; Balasubramanian et al., 1994]. a[…] Hence, a reduction in profilin or tropomyosin function decreases For3 levels in a Sty1-dependent fashion (i.e. in the absence of LatA treatment), further revealing the general significance of the SAPK-For3 branch during the control of actin cables and CAR integrity in response to stress.”

The modified Discussion section also reads:

“Moreover, actin cable and CAR perturbation induced by temperature-sensitive alleles of profilin and tropomyosin, without LatA treatment, were accompanied by downregulation of For3 levels that once more was not observed in the absence of Sty1, thus reinforcing the biological relevance of the SAPK pathway as a negative regulator of this formin in response to CAR damage.”

3) The resistance of sty1 mutants to low doses of LatA raises a simple possibility that LatA does not enter the sty1Δ cell as readily as in WT cells. This could, for instance, be due to potential differences in the cell wall. If true, it would be a major caveat. One suggested test may be to assay dose dependent responses of LatA on actin patches, for instance.

To our knowledge, *sty1∆* cells have not been described in the databases and literature (https://www.pombase.org/gene/SPAC24B11.06c), or isolated in genome-wide screens as showing altered cell wall composition and/or integrity (see, for instance Zhou et al. (2013). PLoS ONE 8(5):e65904). Most important, our finding that increased LatA resistance in *sty1∆* cells is suppressed by deletion of a formin (For3), whose assigned biological role is essentially limited to actin cable formation for growth polarity and cytokinesis, and does not have a noticeable role in cell wall integrity, further reinforces this view.

Nevertheless, following the editor and reviewer´s recommendations, we have analyzed the integrity of F-actin patches in wild type versus *sty1∆* cells in the presence of 20 µM LatA. As can now be seen as novel Figure 2—figure supplement 3 in the revised manuscript, the dynamics of disappearance of actin patches with LatA was very similar in wild type vs. *sty1∆* cells, suggesting that LatA-resistance in this mutant is not related to differences in drug entry induced by MAPK deletion.

We have changed the text describing this experiment in the Results section of the revised manuscript as follows:

“The time taken for complete depolymerization of F-actin patches with high concentrations of LatA (20µM) was very similar in wild type and *sty1∆* cells (Figure 2—figure supplement 3), suggesting that LatA-resistance in this mutant is not related to differences in drug entry induced by MAPK deletion”.

Revisions expected in follow-up work:1) A number of quantitation experiments have been done with asynchronously growing cells (Figures 2B, 3E for example, and there are more). Some are backed up by time-lapse imaging with SPB markers, and others not. This needs to be looked at carefully throughout the work to make sure the phenotypes are neither a overestimate or underestimate.

During time-lapse experiments the extent of ring damage during LatA treatment was determined by quantifying the percentage of cells where condensation of cytokinetic nodes after SPB separation is partially or totally blocked, because we observed that ring constriction proceeds normally, even with LatA present, in cells were a mature CAR is already formed. In asynchronous culture experiments ring damage was scored from multiple micrographs taken at different time points after LatA treatment, and is expressed as the percentage of cells with rings showing morphologically altered, and/or aberrantly condensed rings. These criteria have been strictly followed in all the experiments throughout the work. This information, together with a detailed description of the observed phenotypes is included in the revised manuscript in the “Microscopy analysis” section in Materials and methods:

“The extent of ring damage during LatA treatment was determined in time-lapse experiments by quantifying the percentage of cells in which CAR condensation from cytokinetic nodes is strongly compromised or blocked. In asynchronous culture experiments CAR damage was expressed as the percentage of total cells with rings showing morphologically altered, and/or aberrantly condensed rings. “

We believe that a sum of factors, including variations in strains genotypes, the possible soaking up of LatA by the Ibidi chambers, and the mode of scoring CAR damage, might be behind the differences in the quantification of LatA-induced CAR damage in asynchronous vs. time-lapse experiments. We measured the CAR damage in wild type cells treated with 0.2 µM LatA by using either polymer- (employed through this work) or glass-made ibidi chambers, but did not find significant differences in the percentage of damaged rings. However, the possibility that the soybean lectin used to coat the chambers is responsible for soaking up some of the added LatA cannot be ruled out. Moreover, since the CAR status is estimated in asynchronous cultures from micrographs taken at specific time points, it is possible that in some cells the seemingly damaged CARs might still be able to assemble and constrict. The possible explanations have been included in the Results section of the revised manuscript as follows:

“The higher percentages of cells with damaged CAR found during LatA treatment in asynchronous cultures as compared to the time-lapse experiments may be due to a combination of factors, including genotypic differences among strains, soaking up of small LatA concentrations by the chambers during time-lapse observations, and the different scoring method, since in asynchronous cultures cells with seemingly damaged CARs might still be able to assemble and constrict.”

2) It is not clear how the authors explain the fact that the drop in for3 levels happens very slowly, whereas the phenotypic defects happen in minute timescales. It will be informative to hold the cell cycle using a cdc25-22 or nda3-KM311 and investigate the reduction and increase of for3 levels upon LatA treatment. It is important to know whether the cytokinesis phenotype is a comorbidity or the main effect of LatA induced reduction in for3 levels.

We have thoroughly explored the effect of low LatA concentrations (0.2 µM) on For3 levels in wild type cells within a shorter time-scale (up to 1 hour of incubation). In these conditions, only in two out of eight biological replicates we observed a reduction in For3 levels after 40-60 min of treatment (not shown). However, when the LatA concentration was raised to 1 µM, the reduction in For3 levels became statistically significant after 60 min of treatment.

**Author response image 1. respfig1:** *S. pombe* wild type cells expressing genomic For3-3GFP fusion were grown in YES medium to mid-log phase, and treated with 1μM LatA for the indicated times. Total For3 levels were detected with anti-GFP antibody, Anti-Cdc2 was used as a loading control. Lower panel: quantification of Western blot experiments. Relative For3 levels are represented as mean ± SD (biological triplicates). ***, *P<0.001*; ns, not significant, as calculated by unpaired Student’s t test.

Next, and following the reviewer´s recommendation, we constructed a strain expressing the For3-GFP fusion in an analog-sensitive *cdc2-asM17* background (Aoi et al. (2014) Open Biology 4(7):140063. doi: 10.1098/rsob.140063). Cells from exponentially growing cultures were treated with 1 µM 3-MB-PP1 for 3 h to hold the cycle at G2, and For3 levels were followed at different times after ATP-analog washout in the presence of 0.2 µM LatA that was added 10 min after release from the G2 arrest, when cells are quickly entering into mitosis (~20% binucleated cells) and the CAR is already assembling. As can be seen in the Figure 5G, in these conditions we observed a significant drop in For3 levels as soon as 30 min after treatment, that fairly agree with the high percentage of cells (~80%) that show damaged CAR under these conditions (Figure 2G in the revised manuscript).

Considering the above results as a whole, we believe that the possible decline in For3 protein levels at early times in asynchronic cultures treated with low LatA concentrations lies at the limit of detection of the Western blot assay. Moreover, while this method detects total For3 levels, it does not give information about the specific pools of the formin, which may undergo subtle reductions at defined sub-cellular locations (i.e. in CAR vs. cell poles). The quick decrease in For3 levels observed in synchronized cells at the anaphase onset with the CAR already assembling is in agreement with this prediction. These observations reinforce the notion that reduction of For3 levels is a main factor responsible for the CAR integrity defects observed in the presence of LatA.

We have included new panel G within the Figure 5 of the revised manuscript, together with the following text:

“We did not observe a significant decrease of For3-3GFP levels at early incubation times (0-60 min) in wild type cells treated with low LatA concentrations (0.150-0.200 µM), even though a high percentage of cells (~80%) showed damaged CAR under these conditions (Figure 2G). […] This observation suggests that reduction of For3 levels is an essential factor responsible for the CAR integrity defects in the presence of LatA.”

3) This work would gain in significance if it explored more about the mechanism of how Sty1 pathway affects For3 proteins levels. Could it be regulating translation or stability of the protein? For instance, the authors could test whether Sty1 affects For3 stability using a cycloheximide experiment and/or measure For3 translation.

See our response to Essential revisions for this paper point #1.

4) For3 is best known for its role in actin cable assembly. Given that Sty1 affects For3 levels even in interphase cells, the authors should investigate how stress and Sty1 affect actin cables as well. Do other stresses and the Sty1 pathway impact actin cable assembly, and could this also be suppressed by formin activation or overexpression.

We agree that the putative negative effect of SAPK signalling on cable assembly and/or integrity in fission yeast will surely deserve further insight considering the main role of For3 as a nucleator of actin cables. To this end we have comparatively examined and quantified the percentage of cells with remaining actin cables in wild type versus *sty1∆* cells during unperturbed growth and after treatment with several known stimuli which strongly activate Sty1, including osmotic saline, heat and oxidative stresses. Notably, all of the above stressors prompted a clear reduction in the percentage of cells with remaining cables in wild type cells but not in *sty1∆* cells, suggesting that Sty1-dependent downregulation of actin cable integrity is a general feature in response to environmental changes:

These experiments have been included as a new Figure 2—figure supplement 4 in the revised manuscript, and the corresponding modified text in the Results section now reads as follows:

“Moreover, the percentage of cells with remaining cables became reduced in wild type cells in response to environmental stimuli that activate Sty1, like osmotic saline stress (0.6 M KCl), heat shock (40ºC), or oxidative stress (1 mM H_2_O_2_) [Perez and Cansado, 2010], but not in *sty1∆* cells (Figure 2—figure supplement 4). This suggests that Sty1 activity downregulates actin cable integrity in response to environmental stimuli and actin perturbations induced with LatA. “

The images of actin shown in Figure 2 suggest that actin cables in sty1D cells look brighter; effects on actin cables could be quantitated.

We have explored this possibility by measuring the maximal fluorescence of individual cables (n=100) obtained from maximum projection images of exponentially growing wild type and *sty1∆* cells expressing LifeAct-GFP. As shown in Author response image 2, there were not significant differences in the average fluorescence intensity of cables between both strains. Fluorescence was also highly variable within a given background (WT: 1225 + 508.7; *sty1∆*: 1144 + 507.0) (as mean + SD).

**Author response image 2. respfig2:** 

Is there a reason why stress pathway should inhibit actin cables?

SAPK negative control of actin cables in response to environmental stresses might be required to block polarized growth. This is sustained by previous observations showing that actin cables are required for the transport of Scd1 (Kelly and Nurse (2011). Mol. Biol. Cell. 22(20):3801-11), which is the main GEF that locally activates Cdc42 GTPase for polarized growth (Tay et al. (2018). J. Cell Sci. 131(14):jcs216580). However, we currently lack experimental evidence to support this hypothesis.